# Highly efficient color-tunable organic co-crystals unveiling polymorphism, isomerism, delayed fluorescence for optical waveguides and cell-imaging

Debasish Barman [1], Mari Annadhasan [2], Anil Parsram Bidkar[3], Pachaiyappan Rajamalli[4], Debika Barman [1], Siddhartha Sankar Ghosh [5,6] ✉, Rajadurai Chandrasekar[2] ✉ & Parameswar Krishnan Iyer [1,6] ✉

Photofunctional co-crystal engineering strategies based on donor-acceptor π-conjugated system facilitates expedient molecular packing, consistent morphology, and switchable optical properties, conferring synergic 'structure-property relationship' for optoelectronic and biological functions. In this work, a series of organic co-crystals were formulated using a twisted aromatic hydrocarbon (TAH) donor and three diverse planar acceptors, resulting in color-tunable solid and aggregated state emission via variable packing and through-space charge-transfer interactions. While, adjusting the strength of acceptors, a structural transformation into hybrid stacking modes ultimately results in color-specific polymorphs, a configurational cis-isomer with very high photoluminescence quantum yield. The cis-isomeric co-crystal exhibits triplet-harvesting thermally activated delayed fluorescence (TADF) characteristics, presenting a key discovery in hydrocarbon-based multicomponent systems. Further, 1D-microrod-shaped co-crystal acts as an efficient photon-transducing optical waveguides, and their excellent dispersibility in water endows efficient cellular internalization with bright cell imaging performances. These salient approaches may open more avenues for the design and applications of TAH based co-crystals.

Organic photofunctional materials based on co-crystals, have received immense attention owing to their unique molecular stacking, long-range assembly, high photoluminescence efficiency, and improved semiconducting functionality[1,2]. Co-crystal engineering strategies of single-component building blocks into multicomponent systems displayed remarkable performances in organic solid-state lasers (OSSLs)[3], ambipolar organic field-effect transistors (OFETs)[4], organic light-emitting transistors (OLETs)[5], multi-stimuli-responses (MSR)[6–8], two-photon absorption[9], up-conversion emission[10,11], optical waveguides (OWGs)[12], and bio-imaging[13]. Recognizing the wide use of organic functional co-crystals, many attempts were devoted to develop such co-crystals, noticeably electron-rich donor (D) and electron-deficient

[1]Department of Chemistry, Indian Institute of Technology Guwahati, Guwahati 781039, India. [2]School of Chemistry, and Centre for Nanotechnology University of Hyderabad, Gachibowli, Prof. C. R. Rao Road, Hyderabad 500046, India. [3]Department of Radiology and Biomedical Imaging, University of California, San Francisco, San Francisco, CA 94143, USA. [4]Materials Research Centre, Indian Institute of Science, Bangalore 560012, India. [5]Department of Biosciences and Bioengineering IIT Guwahati, Guwahati, Assam, India. [6]Centre for Nanotechnology, Indian Institute of Technology Guwahati, Guwahati 781039, India. ✉e-mail: sghosh@iitg.ac.in; r.chandrasekar@uohyd.ac.in; pki@iitg.ac.in

acceptor (A) based charge-transfer (CT) co-assemblies were demonstrated to be the best approaches so far[14]. Versatile noncovalent intermolecular interactions (NCIs) viz. hydrogen bonds (HBs), halogen bonds (XBs), π-π stacking, including CT played a pivotal role to define directional interactions, ordered structures, shape/size organization as well as their multifunctional behavior[15]. Specifically, organic CT co-crystals are of potential interest due to their unique arrangements between D-A moieties that benefit from realizing photonic and electronic performances compared to the lone synergistic constituents[12,16]. Besides, different hybrid molecular stacking and definite aggregation patterns such as mixed stack and segregated stack in H- or J-type assembly[17] and color-tunable polymorphs[18] endows significant contribution to their optoelectronic properties[19]. Nevertheless, polymorphs/co-crystals have mainly catered to the pharmaceuticals field[20]. Hence, the evolution of such systems as semiconductors have often been marred by intricate designs or static molecular structures and inefficient optical properties, thereby lacking clarity and failing to describe structure-functions correlation in depth. Nevertheless, the ongoing pursuit of emerging fluorescent materials and harvesting triplets in pure organic molecules is a great challenge. Few CT-active organic emitters have enabled to harvest triplet excitons via an effective rate of reverse intersystem crossing (RISC) process at low splitting energy ($\Delta E_{ST}$), which dictate as thermally activated delayed fluorescence (TADF) and the rate of RISC ($k_{RISC}$) is described in the given Eq. 1[21].

$$k_{RISC} \propto \exp\left(-\frac{\Delta E_{ST}}{k_B T}\right) \qquad (1)$$

The small $\Delta E_{ST}$ between lowest singlet ($S_1$) and triplet ($T_1$) states allowed non-radiative triplet population inversion to regenerate radiative singlet with the aid of thermal energy and delayed fluorescence from $S_1$ to $S_0$. A typical TADF emitter designed by direct covalent bonding at orthogonally linked D-A moieties offered very high quantum efficiency, longer lifetime and improved device performances, thus, receiving significant interest in OLEDs[22]. However, the existence of TADF-co-crystals are seldom found in literature[23], presumably due to either the lack of insights into mostly reported co-crystal design or the planar co-formers remaining ineffective to fulfill the stringent requirement for TADF process. Unfortunately, most of the reported co-crystal structures were constructed by using highly planar and rigid constituents, and they showed weak intermolecular interactions and

notorious aggregation-caused quenching (ACQ) behavior during co-crystallization[24].

To overcome these issues, twisted aromatic hydrocarbons (TAHs) can play a crucial role, as they hold high electron-rich π-conjugated non-planar molecular structures, exhibiting strong electron-donating ability and tunable energy band that demonstrates fascinating optoelectronic properties[25]. Hence, TAHs-based donor is favorable to newer photo functionality to a greater extent as they possess inherent aggregation-induced emission (AIE) behavior via restriction in rotation (RIR) in solid state[26]. Moreover, the dynamic nature of TAHs can easily be deformed by external factors such as solvent, temperature, diverse crystal packing, interactive CT-strength at different crystallization environments, encouraging to achieve aggregation-induced enhanced emission (AIEE) in the co-crystals of polymorphs and configurational isomer, providing newer avenues with unusual optical properties[17,27]. Hence, if the combined advantages of two major co-crystal engineering strategies of polymorphism and configurational isomerism were synergized, the route and design to achieve unique photo-functional materials would be broadened[28,29]. Hence, the rational design of TAH-based multifunctional co-crystals with improved luminescence in aggregated/solid states, and promising photon harvesting characteristics, is of great significance and remains an imperative research area so far.

In this work, we successfully formulate a series of four TAH-based luminescent CT-co-crystals using simple building blocks, referred to as TAHOFN, TAHTFPN_G, TAHTFPN_O, and TAHTCNB, respectively. All the integrated CT-complexes display different luminescence behavior in the presence of different strengths of acceptors, as shown in Fig. 1. Interestingly, TAHTFPN-G and TAHTFPN-O are the color-specific polymorphs, and stable in trans-isomer form, while TAHTCNB exhibits unusual triplet harvesting configurational cis-isomer. Experimental and theoretical results suggest that all the integrated co-crystals exhibit tunable energy band gap ($E_g$), variable degree of CT and multiple NCIs between aromatic cores, conveying defined packing with directionality and specificity to their self-assembly structures, thereby giving bright luminescence ($\phi_F$ = ~77%) and show efficient wave-guide performances.

Remarkably, these co-crystals exhibit different stacking arrangements (H-type mixed stack and J- type segregated stack aggregation packing), resulting in markedly different optical properties in solid-state[15]. The very high rigidity in hindered cis-geometry leads to narrow $E_g$ and $\Delta E_{ST}$ for TAHTCNB co-crystals, hence, exhibiting strong red

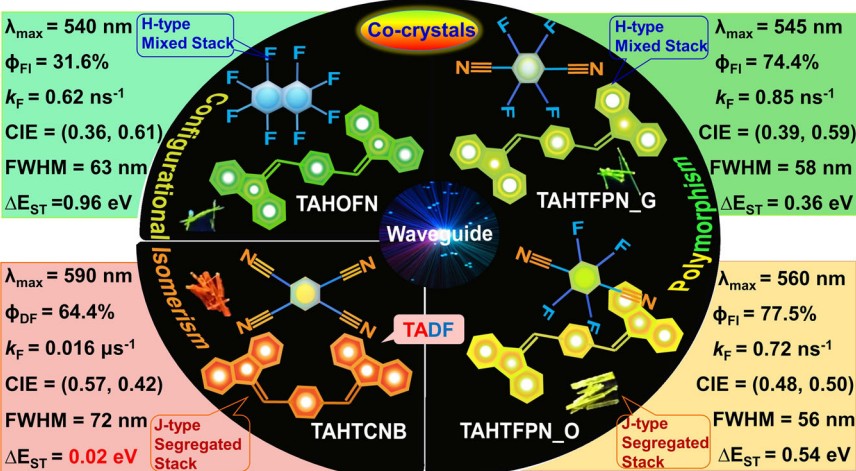

**Fig. 1 | Schematic illustration of luminescent co-crystals.** TAHOFN, TAHTFPN_G, TAHTFPN_O and TAHTCNB composed of twisted aromatic hydrocarbon donor (TAH) and planar acceptors octafluoronapthalene (OFN), tetra-fluoropterepthonatonitrile (TFPN), and tetracyanobenzene (TCNB) with their respective tunable packing-oriented color-tunable polymorphs and configurational isomeric binary assembly structures.

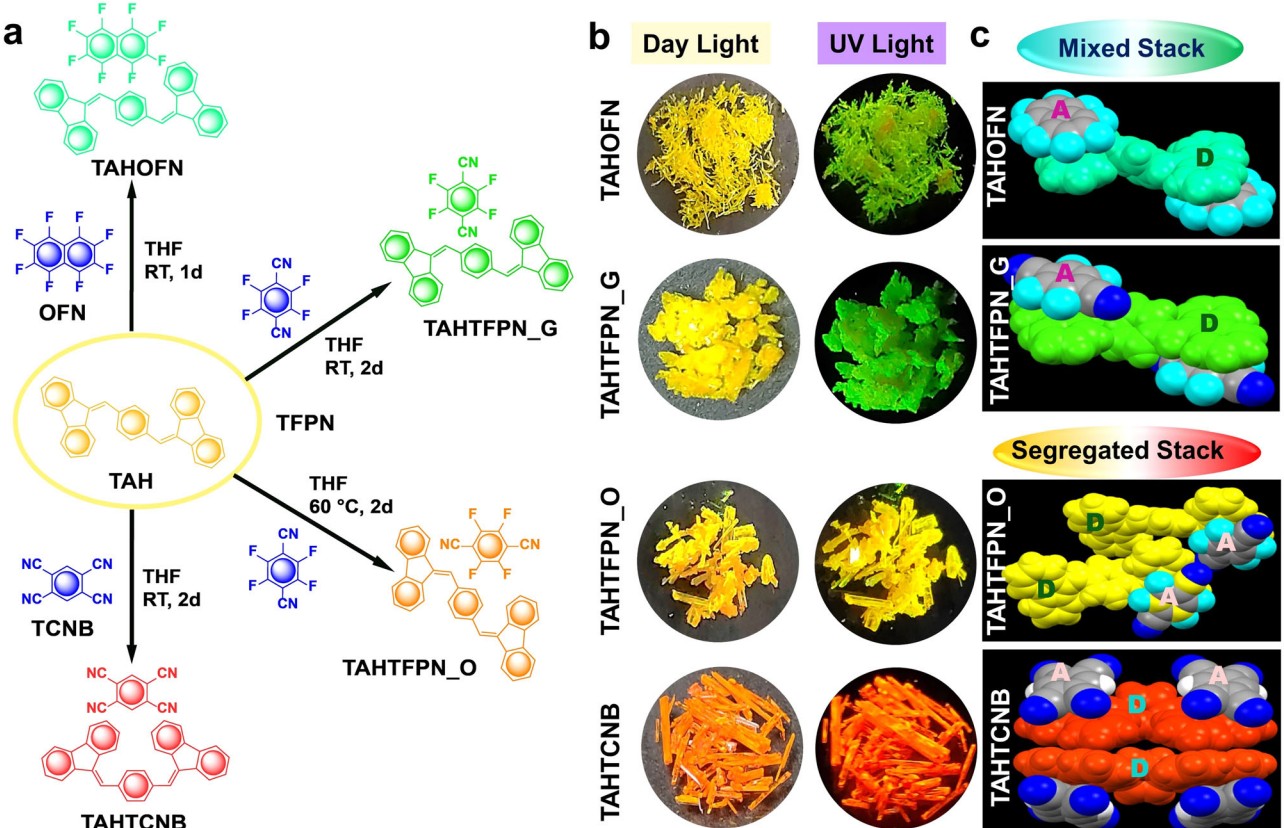

**Fig. 2 | Preparation and physical properties. a** Route of formulation for the co-crystals TAHOFN, TAHTFPN_O, TAHTFPN_G, and TAHTCNB. **b** Images of prepared co-crystals captured under 365 nm UV-lamp off and on condition. **c** Two different stacking modes of TAHOFN, TAHTFPN_G, TAHTFPN_O, and TAHTCNB.

emission (~600 nm) as well as triplet harvesting TADF-characteristics, a newer avenue for purely hydrocarbon-based TADF-co-crystals design. Uniform chain-like molecular arrangements in co-crystals through continuous π-π interactions result in highly rigid crystalline structures, rationalizing the unidirectional photon propagation at a very low optical loss coefficient. Besides, the TAH co-crystals exhibit an unusual AIEE phenomenon with good water dispersibility, endowing very rare cellular internalization with bright cell imaging performances. This work thus presents very remarkable multifunctional properties, utilizing the different co-assembly approaches of TAH resulting in tunable emission properties, including mechanistic insights and photon harvesting characteristics. Consequently, this work not only confirms that constructing polymorphic and configurational isomeric co-crystals can be an effective way to design synergistic luminescent CT-co-crystals but also presents a clear understanding of the relationship between variant co-assembly designs, aggregation-induced enhanced emission, color-tunability and self-guiding singlet-triplet optical waveguides and biological/therapeutic applications.

## Results

### Co-crystals design and formulation

Four color-tunable fluorescent rod-shaped single-crystalline co-crystals denoted as TAHOFN, TAHTFPN_G, TAHTFPN_O, and TAHTCNB were prepared via solvent evaporation method, by using 1:1 stoichiometry ratio of the precursor molecules in THF solvent. While TAH, a dibenzofulvene derivative, was chosen as an electron-rich twisted non-planar donor, octafluoronaphthalene-96% (OFN), tetrafluoropterepthanitrile-99% (TFPN) and 1,2,4,5-Tetracyanobenzene-97% (TCNB) cores were chosen as an electron-deficient planar acceptor. TAH promptly co-assembled with the selected acceptors viz. OFN, TFPN, and TCNB (within 48 hours) via solvent evaporation, whereas very swift (30 secs)

co-crystallization was also observed via solid-state grinding (SSG) methods (shown in Supplementary Movie 1, Supplementary Movie 2, and Supplementary Movie 3 respectively). However, evaporation method resulted in rod-like four distinct colored CT complexes, that were used for all the structural and photophysical characterization. Especially, color-specific polymorphic co-crystals were obtained by controlling the temperature while dissolving the binary mixtures. TAHTFPN_G was obtained at room temperature (RT), whereas TAHTFPN_O was obtained at 60 °C. In contrast, others co-crystals were grown at RT and no other polymorphs or other structures were observed under temperature variation (Fig. 2a). Notably, we could easily tune the excited state properties by modulating the different acceptor strengths. At the same time, geometrical isomerism further assists in harvesting triplets and exhibiting TADF behavior. Co-crystal structures and distinct packing modes via supramolecular interactions were analyzed by single-crystal X-ray diffraction crystallography method. Rod-like single crystals were grown with the two most common packing mode viz. H-type mixed stacking exhibiting -D-A-D-A- columns with face-to-face π-π overlapping for off-green emissive TAHOFN and bright green emissive TAHTFPN_G, whereas, J-type segregated stacking mode showed an appropriate slippage of face-to-face overlapping in separated -D-D- and -A-A- columns for the yellow emissive TAHTFPN_O and red emissive TAHTCNB (Fig. 2b, c). Luminescence characteristics of the carefully prepared single co-crystals were recorded via fluorescence microscopy, and the images were collected under 405 nm excitation (Supplementary Fig. 1). These differences in mixed stack and segregated stack packing modes resulted in distinct optical properties of the co-crystals.

In principle, in the selected π-acceptor aromatic cores, the density of aromatic π-clouds at naphthalene in OFN is higher than phenyl cores in TFPN and TCNB. Besides, the strong electron-withdrawing behavior

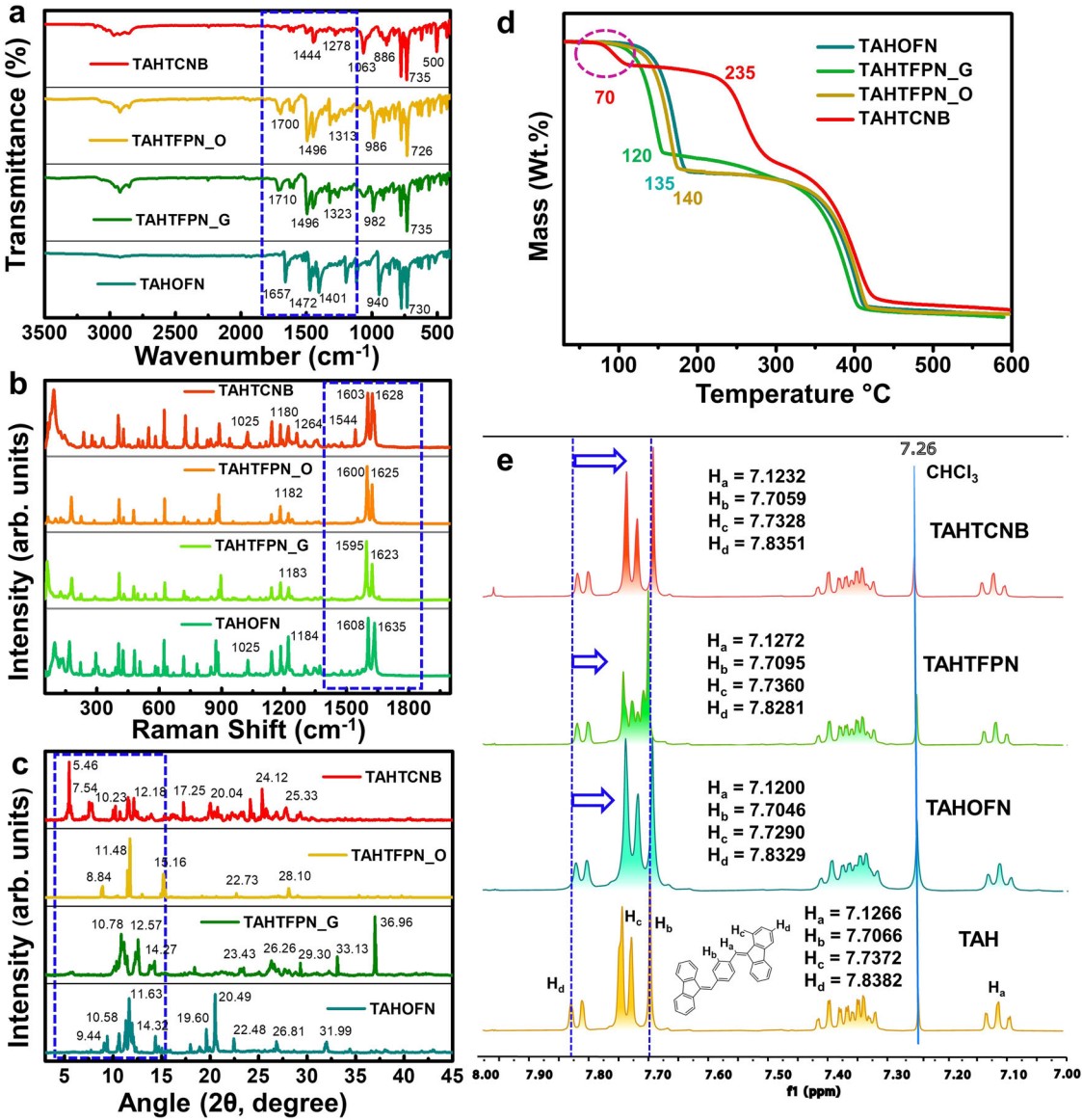

**Fig. 3 | Characterization of co-crystals. a** FTIR spectra, **b** Raman spectra, **c** PXRD patterns, and **d** TGA graph for the co-crystals of TAHOFN (off-green line), TAHTFPN_G (green line), TAHTFPN_O (orange line), and TAHTCNB (red line) respectively. **e** Stacked [1]H NMR spectra for TAHOFN, TAHTFPN, and TAHTCNB co-crystals, recorded in CDCl₃ at 298 K. (The blue dotted boxes in the inset of a, b, c is representing the main difference in the frequencies and peaks patterns for the co-crystals.

of −CN group in TCNB diffusesing the phenyl π-clouds to −CN rather than poor electron withdrawing −F, thus, less likely available for strong orthogonal π-π stacking. Therefore, a face to face π-π stacking became possible between TAH and OFN and formed mixed stack as confirmed by their SC-crystal structures. Likewise, TAH and TFPN favors some extent of orthogonal π-π stacking at RT, however, during higher temperature crystallization, the density of π-clouds in phenyl is less available and formed segregated stack TAHTFPN_O as similar stacking fashion to TAHTCNB.

### Characterization of Co-crystals

Fourier transform infrared (FTIR) spectroscopy for all the co-crystals were analyzed to prove the formation of CT complexes (co-crystals), characterizing the π-π stacking, and intermolecular HBs and CT-interactions between D and A. In all the co-crystals the complete involvement of both co-formers was observed with a certain shift of their stretching frequencies (Supplementary Figs. 2–5). However, all the co-crystals showed a significant shift in stretching frequencies due

to their different functional groups such as −F and −CN, respectively (Fig. 3a). Furthermore, micro-Raman spectroscopy (at 633 nm) demonstrated an entirely different peak compared to the co-former signals (Supplementary Figs. 6–9).

In addition, a new peak was observed at 1025 cm⁻¹ in TAHOFN and 898 cm⁻¹ in TAHTFPN_G, while multiple new frequencies of 552 cm⁻¹, 727 cm⁻¹, 1025 cm⁻¹, 1260 cm⁻¹ were seen in TAHTCNB, which caused the variation in CT and HB interactions on the co-assembly formation (Fig. 3b). More importantly, the appearance of newer peaks in the co-crystals clearly depicted the presence of intermolecular interactions[25]. In contrast, powder X-ray diffraction (PXRD) confirmed the co-crystallization of the twisted TAH system in the presence and absence of planar OFN, TFPN, and TCNB (Supplementary Figs. 10–13). Compared to pure precursor molecules, several new characteristic peaks appeared in TAHOFN co-crystals, depicting the new molecular arrangement upon co-assembly formation. Although TAHTFPN_G and TAHTFPN_O co-crystals were composed of similar co-formers, the different diffraction patterns suggested a packing difference and

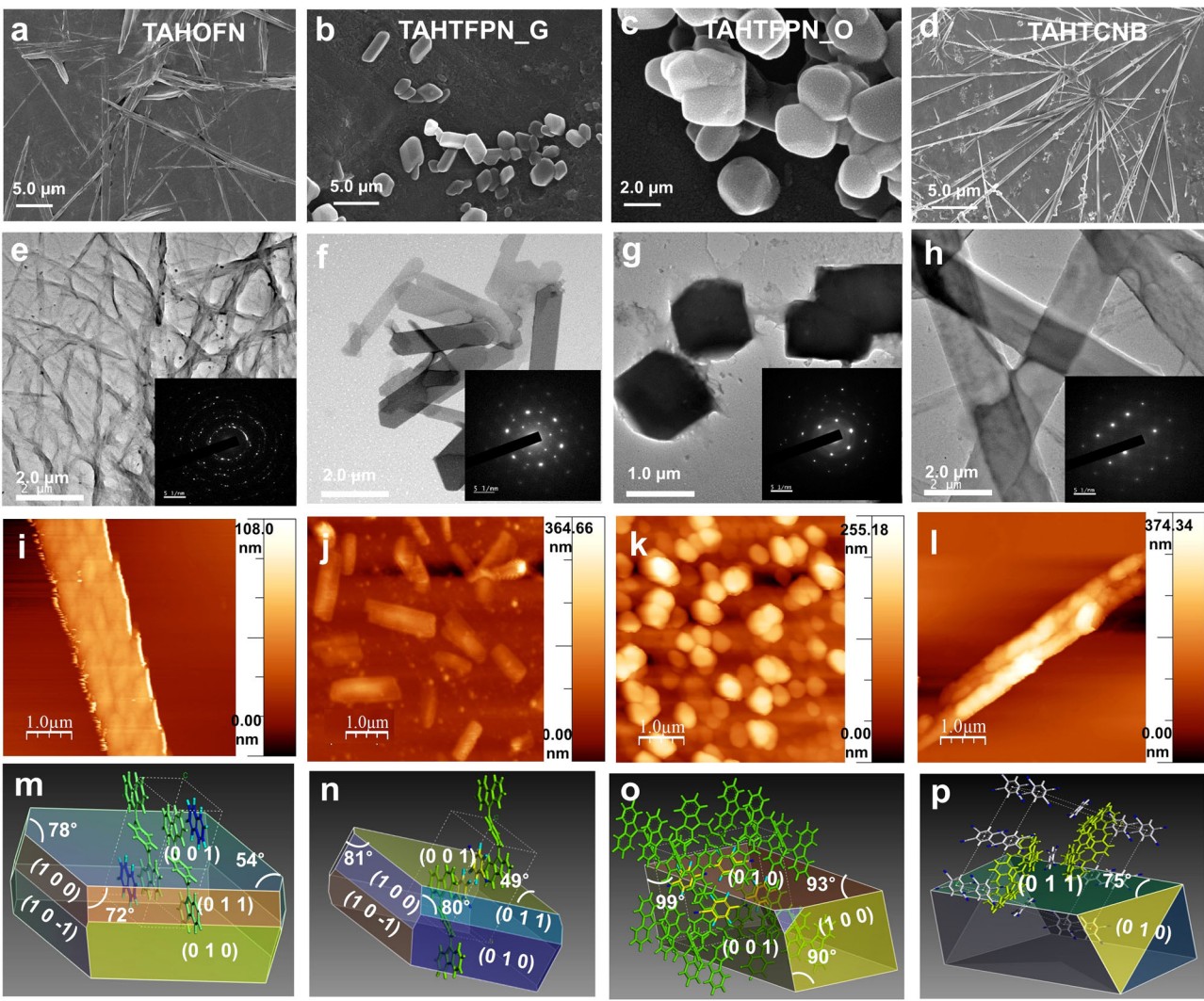

**Fig. 4 | Microcrystals size and morphology visualization. a–d** Low-magnification (scale bar: 2 & 5 µm) FESEM images. **e–h** Low-magnification TEM images (scale bar 2 µm) of crystals (inset SAED pattern collected for crystallinity). **i–l** AFM images with height profile (1 µm scale bar). **m–p** Growth morphologies were anticipated by Materials studio 17 software for all the co-crystals of TAHOFN, TAHTFPN_G, TAHTFPN_O, and TAHTCNB single crystals geometry based on the calculated attachment energies calculations, respectively.

polymorphic behavior (Fig. 3c)[30]. Further, the thermal stability of the individual precursors and co-crystals studied by thermogravimetric analysis (TGA) demonstrated the appearance of new peaks for co-crystals with two thermal degradations at 135 °C, 360 °C for TAHOFN, 120 °C, 360 °C TAHTFPN_G, 140 °C, 360 °C for TAHTFPN_O and 235 °C, 360 °C for TAHTCNB (Fig. 3d and Supplementary Figs. 14–17). TAHTCNB co-crystal starts to decompose initially at around 70 °C (shown in Fig. 2d, inset: red circle) and this degradation is due to the presence of solvent molecule (presumably THF) in the crystal lattice. However, actual decomposition temperature of TAHTCNB co-crystal was noticed to be round 235 °C, revealing high thermal stability of TAHTCNB rather than the three other co-crystals. Between the two polymorphs, TAHTFPN_O showed more thermal stability than TAHTFPN_G. These findings also suggested that co-crystals with the more thermally stable TAH donor (360 °C) produced a new packing mode that improves the thermal stability relative to pure constituents. Furthermore, differential scanning calorimetry (DSC) study showed an endothermic peak at 211 °C, 220 °C, 190 °C, 196 °C for TAH, TAHOFN, TAHTFPN_G, and TAHTFPN_O, while TAHTCNB exhibited multiple peaks at 107 °C, 207 °C, 217 °C and 263 °C respectively (Supplementary Fig. 18). DSC results corroborated the melting point of the respective co-crystals. Interestingly, TAHTCNB presents various other peaks,

which are presumably due to isomeric transition and the presence of some solvent molecule in the crystal lattice. Moreover, the dissimilar melting transition for all the co-crystals is due to their unique crystal lattice structures in the solid state and associated different multiple noncovalent and electrostatic Coulomb interactions, respectively. Furthermore, [1]H NMR studies demonstrated a consistent up fielded $\delta_{ppm}$ shift on moving from TAHOFN to TAHTCNB co-crystals suggesting strong electron-withdrawing nature of acceptors shielding the protons of TAH (Fig. 3e and Supplementary Fig. 19). Collectively, all these results confirm the successful formulation of co-crystals with different stacking and rigid packing between electron-rich TAH and electron-deficient OFN, TFPN and TCNB, respectively.

## Microcrystals growth morphology visualization

Multiple weak interactions lead to structural transformation and produce polymorphs or geometrical isomers responsible for forming different morphologies and optical properties[31,32]. Therefore, in order to visualize controlled and regular 1D/2D morphology of the co-crystals, low-magnification (scale bar: 5.0 µm and 2.0 µm) FESEM images (Fig. 4a–d) manifested uniform rod shape of the TAHOFN and 2D sheet-like for TAHTFPN_G crystals with size ($D$ = 0.6, 1.15 µm, $L$ = 12, 3.7 µm, $W$ = 0.42, 0.5 µm) and relatively small size prism-shaped and

rod-like 1D morphology of TAHTFPN_O and TAHTCNB with size ($D$ = 1.8, 0.6 µm, $L$ = 2.8, 20.0 µm, $W$ = 0.35, 0.2 µm).

Low-magnification TEM images (scale bar 2 µm) also consisted of the co-assemblies' regular 2D and 1D shapes (Fig. 3e–h). However, a defined diffraction pattern was observed in selected area electron diffraction (SAED) analysis, attributed to the high crystallinity of the co-assembly structures (inset Fig. 3e–h). Further, to visualize the surface morphology, AFM images have also been recorded (scale bar 1 µm) and the afforded images are akin to the FESEM and FETEM images (Fig. 4i–l). The height and RMS roughness were found to be 323.0, 100.3, 101.9, 122.82 nm and 158.75, 22.93, 36.37, 35.37 nm, for TAHOFN, TAHTFPN_G, TAHTFPN_O, and TAHTCNB respectively. Moreover, uniform supramolecular π-stacked co-assemblies through anisotropic interactions lead to perfectly orthogonal/parallel assembly, benefiting to obtained smooth surface topography in 2D block-shaped and 1D-sheet/rod-shaped crystal growth. Further, growth morphologies of the co-crystals were simulated and predicted by using Bravais-Friedel-Donnay-Harker (BFDH) theory and attachment energy principle in Material Studio software[33]. Simulated morphology provides multiple fast-growing crystal facets of {0 0 1}, {0 1 0}, {0 1 1}, {1 0 0}, and {1 0 −1} for both TAHOFN and TAHTFPN_G, respectively (Fig. 4m, n and Supplementary Table 1, 2). Subsequently, the crystal facets of {0 0 1}, {0 1 0}, {1 0 0}, and {1 1 1} for TAHTFPN_O, while for TAHTCNB, only two crystal facets {0 1 1}, {1 0 1} was predicted (Fig. 4o, p and Supplementary Tables 3 and 4). Therefore, TAHOFN and TAHTFPN_G grows into rod and 2D sheet-like crystal in the preferential growth direction at {0 0 1}, where TAHTFPN_O and TAHTCNB grows into cubic prism and rod-shaped crystal in the {0 1 0} and {0 1 1} direction thereby facilitating easy light transmission feature along the major growth direction, respectively. Remarkably, simulated morphology consistent with the experimental observation is anticipated to be regular prism and 1D sheet/rod-like structures with the length direction along c axis. Overall photophysical and optoelectronic properties are largely affected by the size and morphology of materials[34,35]. Thus, modulating molecular packing with desirable morphology via anisotropic interactions in π-conjugated co-assembly is important for the fundamental study and strongly impacts optoelectronic functions.

## Crystal structure analysis

Supramolecular aggregated structures and molecular packing greatly influence the optical properties of π-conjugated materials in solid-state and confirm unique structure-function correlations[36,37]. Therefore, to clearly understand the mode of packing-induced color-tunable fluorescent behavior at their atomic level, SC-XRD structures were solved and analyzed in detail for all four integrated co-crystals. The SC-XRD structures of co-crystal showed three triclinic and space groups of *P-1* for TAHOFN (CCDC NO 2109333), TAHTFPN-G (CCDC NO 2109336) and TAHTFPN-O (CCDC NO 2109335), whereas, TAHTCNB exhibited an orthorhombic crystal system and *Pnmm* a space group (CCDC NO 2109337). Detailed crystal data collection and co-crystal refinement parameters are summarized in Supplementary Table 5. In the SC structures of co-crystals, each donor molecule is connected with adjacent acceptor molecules via multiple intra- and inter-molecular interactions. Face-to-face orthogonal stacking between TAH and OFN aromatic cores was observed for TAHOFN with an angle of 88.7° and intermolecular HBs ($d_1$ =C-F···H-C, 2.54 Å) were observed between π-stacked OFN, where F atom in OFN linked to adjacent H atom present in bridge phenyl ring of TAH donor (Fig. 5a). Subsequently, multiple intramolecular CH-π interactions ($d_2$, $d_3$ $d_4$ = 2.56 Å, 2.37 2.83 Å) with a dihedral angle of 50.10° was observed between dibenzofulvene (DBF) and bridge phenyl ring in TAH donor. Similarly, TAHTFPN_G displayed an orthogonal stacking angle of 80.10° with intermolecular HB ($d_1$ =C-N···H-C, 2.35 Å) and π-π stacking interaction ($d_2$ = 2.39 Å) between TFPN and TAH. Intramolecular interaction was also seen as CH-π

interactions ($d_3$, $d_4$ 2.74 Å, 2.81 Å) with a dihedral angle of 40.92° in TAH donor (Fig. 5b). Moreover, the polymorphic crystal structure of TAHTFPN_O showed slightly staggered stacking between TAH and TFPN with an angle of 76.91°, while two HBs ($d_1$ = C-F···H-C, 2.60 Å and $d_2$ = C-N···H-C, 2.58 Å) and two π-π stacking ($d_3$, $d_4$ = 3.37 Å, 3.40 Å) interactions were realized. Likewise, close intramolecular CH-π interactions ($d_1$, $d_2$ = 2.55 Å, 2.72 Å) with a low dihedral angle 39.30° was also apparent in TAH donor (Fig. 5c). The polymorphs of TAHTFPN_G and TAHTFPN_O are specifically distinct by their inter-molecular NCIs. TAHTFPN_O demonstrated to be a more segregated packing structure than mixed-stacked TAHTFPN_G, resulting in distinguishable optical behavior[38,39]. TAHTCNB displayed intense staggered stacking between TAH and TCNB with an angle of 76.03° and intermolecular HB and π-π stacking ($d_1$ = C-N···H-C, 2.59 Å and $d_2$ = C$_{π-π}$, 3.83 Å) interactions. The cis-conformation of TAH was stabilized by intramolecular CH-π interactions ($d_3$, $d_4$ = 2.82 Å, 2.83 Å) with a dihedral angle of 46.10° between the two planes of DBF and central phenyl planes in TAH (Fig. 5d). Also, the bulk crystal packing demonstrates a mixed-stack packing mode for TAHOFN and TAHTFPN-G, along a-axis into alternating -A-D-A-D- based symmetrical alignment with a very close contact distance of 3.40 Å and 3.54 Å, respectively (Fig. 5e, f). Meanwhile, a segregated stack packing mode along a- and b-axis was seen for TAHTFPN-O and TAHTCNB between TAH and TFPN/TCNB stacked on each other relatively at far distance of 3.65 Å and 3.73 Å, individually in a -DD-AA- based unsymmetrical fashion (Fig. 5g, h). Dynamic nature of TAH core in all the crystals was confirmed from their different twisted angle of 56.87°, 38.90°, 45.18° and 52.51° for TAHOFN, TAHTFPN_G, TAHTFPN_O, and TAHTCNB between central phenyl core and planar DBF core respectively (Supplementary Fig. 20). Thus, a strong dimer was stabilized through two intermolecular H-bonds and π-π stacking that resulted in molecular chain-like assemblies with layer-by-layer arrangement where D and A molecules are interacting via different face to face or edge-to-edge stacking (Fig. 5i–l). The dynamic nature of TAH in the co-crystals and non-bonding interactions are shown and summarized in Supplementary Figs. 21–22 and Supplementary Table 6. Further, electrostatic potential (ESP) mapping, corresponding two-dimensional (2D) fingerprint plots with their counter pitch angle (88.7°, 80.10°, 76.91°, and 76.03°) and static dipole moment (SDM) values of 0.80D, 1.31D, 1.56D, and 1.62D, respectively further justifying favorable red-shifted emission (Supplementary Fig. 23-27)[40]. Therefore, the SC-XRD analysis clarified the abundance of multiple inter-molecular interactions and enabled the formation of distinct and highly ordered packing modes, which confer tunable optical behaviors of the co-crystals. Moreover, rigid and dense sheet and wave-like network for the co-crystals benefits to restrict vibrational relaxation and prevents excitons quenching[32,41]. The low PLQY of TAHOFN can be assigned to the formation of local H-aggregates between TAH and OFN molecules, which leads to a high non-radiative deactivation rate. The presence of plenty of large intermolecular interactions and staggered orientation reduces the fluorescence quenching effect, resulting in high PLQY and revealing clear insights of structure-function correlation which directs into light propagating waveguide applications.

## Optical properties of co-crystals

To investigate the luminescent features of the afforded co-crystals, UV-visible and photoluminescence (PL) studies were performed in bulk crystals and micro/nano-scale crystals respectively. TAHOFN, TAHTFPN_G exhibited blue-shifted absorption at 460 and 430 nm whereas TAHTFPN_O and TAHTCNB showed a red-shifted absorption at 490 and 500 nm related to the absorption of parent TAH (485 nm), deciphering through-space charge transfer (TSCT) interactions between constituent precursors (Supplementary Fig. 28). Previous report on the solid and aggregated state photophysical and morphological properties of TAH donor molecule (Supplementary Fig. 29)[42], provided key evidence to initiate investigations on the aggregation

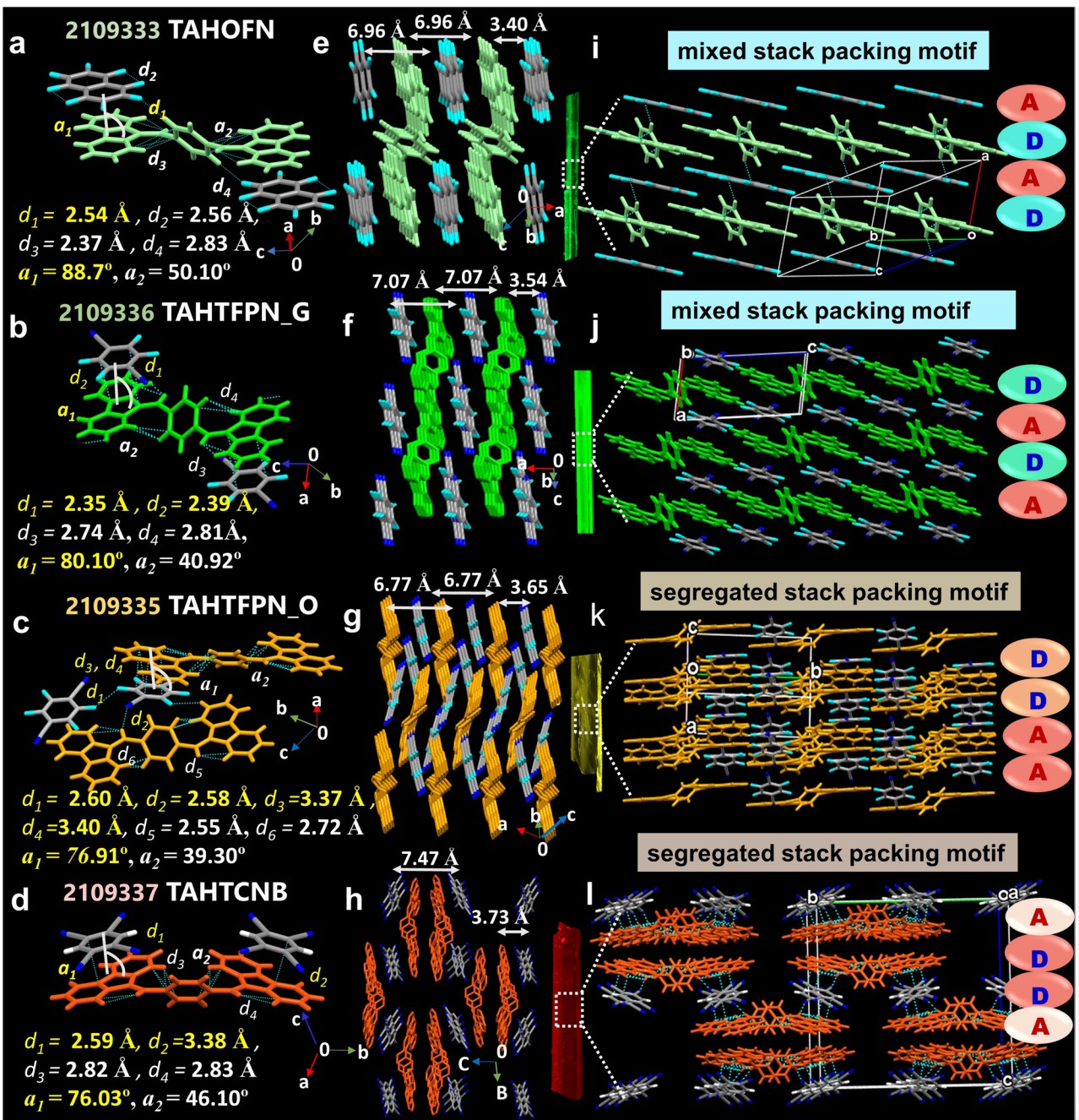

**Fig. 5 | Single crystal X-ray diffraction (SC-XRD) analysis. a–d** SC-XRD structure of co-crystals in unit cell and interactive noncovalent bonding distances and angles at intra and intermolecular fashion. **e–h** Top view of for all four co-crystals at their bulk stacked architectures and relative distances from each constituent. **i–l** Side view in bulk packing pattern and representation of mixed stack TAHOFN and TAHTFPN-G and segregated stack TAHTFPN_O and TAHTCNB co-crystals respectively (inset: fluorescence single crystal image).

behavior of the afforded TAH co-crystals. Hence, initially, AIE behavior was studied using various fractions of THF/water mixtures. Interestingly, co-crystals were found to be non-emissive in THF solution. In contrast, with increasing water fraction ($f_w$) a significant fluorescent enhancement was observed in 90% $f_w$ for TAHOFN at $\lambda_{max} = 505$ nm, and 99% $f_w$ for TAHTFPN_G and TAHTCNB at $\lambda_{max} = 525$ and 595 nm, respectively (Fig. 6a–c). In Fig. 6d–f, the plot of relative PL intensity with different $f_w$ and inset images of co-crystals in THF and water (taken under 365 nm UV-lamp) along with laser-induced ($\lambda_{ex} = 405$ nm) fluorescence confocal images (scale: 5μm) clearly depict the sharp fluorescence increments, while the confocal images showed micro-range fluorescence wire, leaf and ribbon-like micro-structures in the

aggregated state. Since biological studies were performed using DMSO as a stock solution, FETEM images, size distribution by dynamic light scattering (DLS) and stability of colloidal dispersion by Zeta potential studies were performed using diluted DMSO/$H_2O$ (1:9) mixture to confirm the colloidal aggregation behavior of the co-crystals (Supplementary Figs. 30–33).

Therefore, the TAH-based co-assembly approach reduces aggregation-caused quenching effect and showed improved condensed state emission, beneficial for optoelectronics and bio-imaging[13,43]. The steady state PL spectra for all co-crystals displayed a broad emission range from 450 to 700 nm, while gradual red-shifted emission peaks were observed mainly located at $\lambda_{max} = 540$,

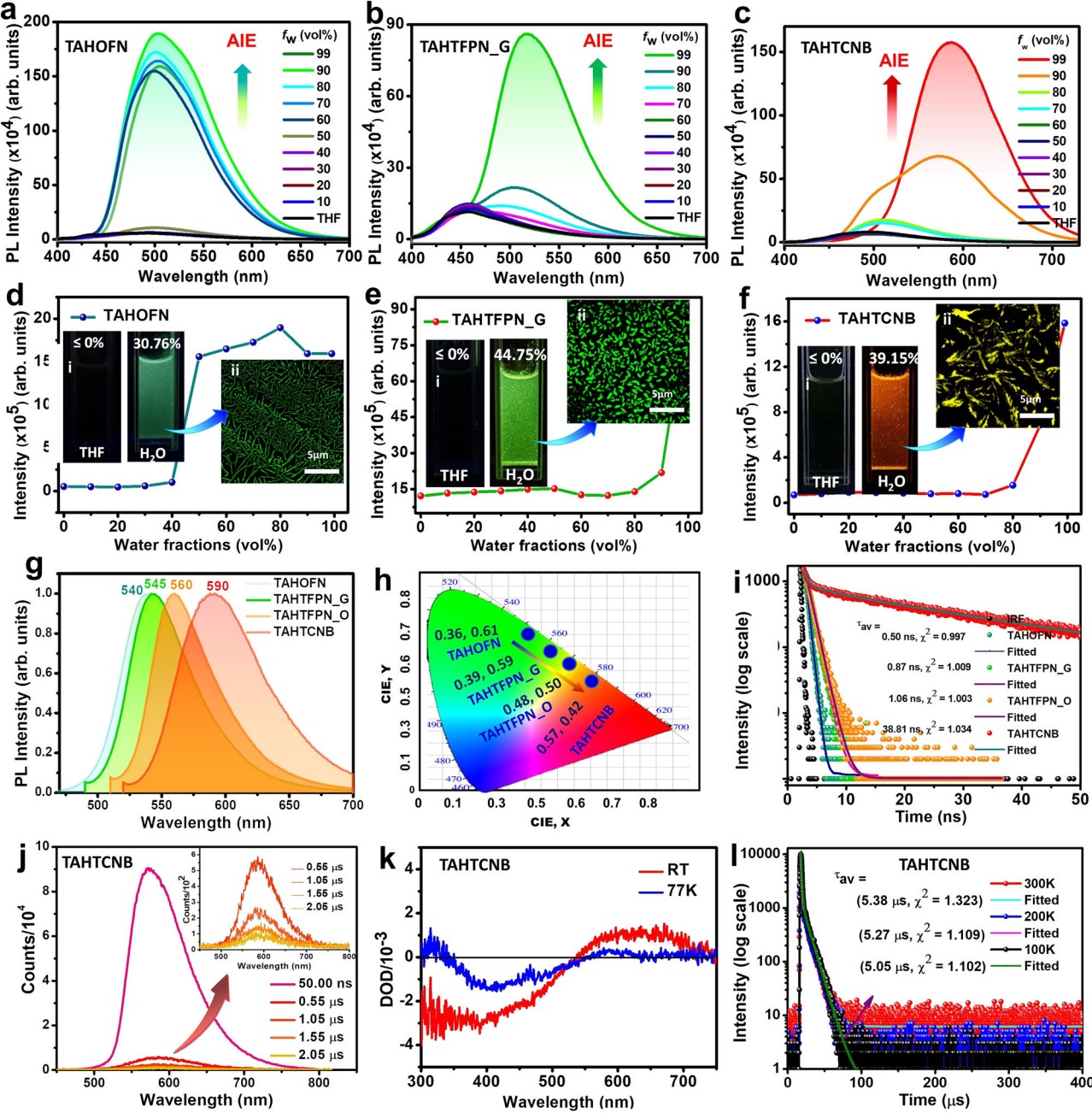

**Fig. 6 | Aggregated and solid state photophysical properties.**
**a–c** Photoluminescence (PL) spectra in various THF/Water mixture fractions.
**d–f** Relative PL intensity plot at different THF/Water fraction, inset: images of co-crystals in THF and water (i), confocal fluorescence images of aggregated co-crystals (ii) TAHOFN, TAHTFPN_G, TAHTCNB. **g** Solid state PL spectra of all the afforded co-crystals in the crystalline forms, TAHOFN, TAHTFPN_G, TAHTFPN_O, and TAHTCNB, respectively. **h** CIE-coordinates for the integrated co-crystals. **i** TRPL spectra in log-scale for all the co-crystals in the solid state. **j** Laser-induced fluorescence (LIF) spectra for TAHTCNB at ns and μs pulse delay for TAHTCNB co-crystal. **k** Transient absorption (TA) spectra at room temperature (RT) and 77 K for TAHTCNB. **l** Transient PL spectra at different temperatures (100–300 K) for TAHTCNB.

545, 560, and 590 nm with full-width at half maxima (FWHM) 63, 58, 56, and 72 nm respectively, corresponding to their variation in through-space CT between respective D-A pair[44] (Fig. 6g). Absolute PL quantum yield (PLQY) has been determined to be 31.6%, 74.4%, 77.5%, and 64.4% for TAHOFN, TAHTFPN_G, TAHTFPN_O, and TAHTCNB, respectively, using an integration sphere (Supplementary Figs. 34–37). To the best of our knowledge, such high PLQY values (77.5%) for co-crystalline materials are rarely reported. Exact emission colors were calculated from the Commission Internationale de L'Eclairage (CIE) chromaticity coordinates values of (0.36, 0.61) for TAHOFN, (0.39, 0.59) for TAHTFPN (0.48, 0.50) for TAHTFPN and

(0.57, 0.42) for TAHTCNB respectively (Fig. 6h). To further understand the in-detail excited state information, the fluorescence decays were recorded at solid crystalline form using TRPL at time-correlated single photon counting (TCSPC) method and the average lifetime ($\tau_{av}$) values of 0.50, 0.87, 1.06, and 38.81 ns were recorded for TAHOFN, TAHTFPN_G, TAHTFPN_O, TAHTCNB, respectively (Fig. 6i). Unlike, single exponential decay profile of former three co-crystals, TAHTCNB exhibited tri-exponential decay due to the presence of prompt and delayed species. Additional details on excited-state lifetime fitting values of the co-crystals of polymorph and conformational isomer are summarized in Supplementary Tables 7–10.

**Table 1 | Summary of photophysical and electronic properties of the co-crystals**

| Co-crystals | $\lambda_{UV}$ (nm) | $\lambda_{PL}$ (nm) | CIE (x, y) | HOMO (eV) | LUMO (eV) | $E_g$ (eV) | $\Delta E_{ST}$ (eV) | $\tau_{av}$ (PF) (ns) | $\tau_{av}$ (DF) (μs) | $\phi_{PL}$ (%) | $K_F$ (ns$^{-1}$)[a] | $\mu$ (D) | $\alpha'$ (dB/μm) |
|---|---|---|---|---|---|---|---|---|---|---|---|---|---|
| TAHOFN | 460 | 540 | 0.36, 0.61 | −5.51 | −2.76 | 2.75 | 0.96 | 0.50 | – | 31.6 | 0.62 | 0.80 | 0.44 |
| TAHOFN_G | 430 | 545 | 0.39, 0.59 | −5.58 | −2.69 | 2.89 | 0.36 | 0.87 | – | 74.4 | 0.85 | 1.31 | 0.0742 |
| TAHOFN_O | 490 | 560 | 0.48, 0.50 | −5.48 | −2.80 | 2.68 | 0.54 | 1.06 | – | 77.5 | 0.72 | 1.56 | – |
| TAHTCNB | 500 | 590 | 0.57, 0.42 | −5.55 | −3.52 | 2.03 | 0.02 | 38.81 | 5.38 (300 K) 5.05 (100 K) | 64.4 | 0.016 | 1.62 | 0.0837 |

[a]Radiation constants [$k_F = \phi/\tau$] calculated from fluorescence QY and prompt fluorescence lifetime of crystals at RT in air.

The radiation constants ($k_F$) of co-crystals were calculated to be 0.62 ns$^{-1}$, 0.85 ns$^{-1}$, 0.72 ns$^{-1}$, and 0.016 ns$^{-1}$ by using the formula $k_F = \phi_{PL}/\tau$, which is a much smaller value than the single component or/and other typical CT crystals[45]. These, small $k_F$ values corroborate, that the fluorescence originated from CT state. To confirm TADF property laser-induced fluorescence (LIF) study was performed by applying nanoseconds (50 ns) and microseconds (0.55, 1.05, 1.55, and 2.05 μs) pulse laser excitation for 0.5 wt% PMMA doped TAHTCNB film (Fig. 6j). Interestingly, a delayed emission spectrum was observed at various μs delay pulse laser excitation, referred as TADF[23]. The triplet harvesting behavior of TAHTCNB was further confirmed by transient absorption (TA) spectra at room temperature (RT) and 77 K, respectively (Fig. 6k). As expected TA spectra depicted a positive signal near 600 nm at RT, while the intensity was found to be higher than that of 77 K. To verify the TADF characteristics of TAHTCNB, temperature-dependent decay profile for delayed lifetime and delayed emission spectra was recorded at 100–300 K temperatures (Figs. 6l, S38 and Supplementary Tables 11–13). In addition, steady state PL (at RT and 77 K) and 77 K phosphorescence spectra was recorded to obtain experimental $\Delta E_{ST}$ value of (0.01 eV), collectively confirming the TADF behavior in TAHTCNB co-crystal (Supplementary Fig. 38). Since, the red emissive TAHTCNB co-crystal experiences a rare intermolecular steric strain due to the cis-configuration adopted by TAH, thereby generating unusual supramolecular architecture, which could suppress the non-radiative relaxation and assist in harvesting triplets via an effective RISC process at reduced $\Delta E_{ST}$ to obtain efficient TADF property[46]. All the optical properties are summarized in Table 1.

**Intrinsic mechanism for color-tunable emission**
Furthermore, the unusual optical properties of co-crystals motivated us to investigate the variable CT/packing interactions more critically and mechanistically to determine the co-crystallization process. Therefore, theoretical simulations were carried out to clearly understand the mechanism of blue/red-shifted absorption/fluorescence and TADF activities, including their dissimilar TSCT behavior in the co-assembled structures by employing time-dependent density functional theory (TD-DFT) calculations from Gaussian 16 programme[47,48]. Frontier molecular orbitals (FMOs) for all the co-crystals were calculated by using the B3LYP/6-31 G(d,p) basis set and level from their single-crystal geometries. The HOMO and LUMO values are calculated; therefore, the related band gap was calculated to be 2.75, 2.89, 2.68, and 2.03 eV for TAHOFN, TAHTFPN_G. TAHTFPN_O and TAHTCNB, respectively (Fig. 7a-i, b-i, c-i, d-i). These values are in good agreement with experimental bandgap values (2.75, 2.35, 2.32, and 2.03 eV) obtained from cyclic voltammetry (CV) (Supplementary Figs. 39–42 and Supplementary Table 14). FMOs distribution suggested that HOMO of the three co-assemblies were mainly located on the donor TAH, whereas LUMO aligned with the acceptors TFPN and TCNB, leading to well-separated HOMO and LUMO, clearly indicating the TSCT characteristics of the co-assemblies[49].

Besides, the energy level for different spin multiplicity states were determined and the $\Delta E_{ST}$ values were calculated to be 0.96, 0.36, 0.54, and 0.02 eV for TAHOFN, TAHTFPN_G, TAHTFPN_O, and TAHTCNB

respectively (Figs. 7a-ii, b-ii, c-ii, d-ii and Supplementary Tables 15–22). Unlike very small $\Delta E_{ST}$ values of TAHTCNB other three co-crystals exhibited large $\Delta E_{ST}$ values (>0.3 eV) and were stable in trans architecture of TAH, which hinders the RISC process and thereby found to be TADF inactive[21]. In Supplementary Table 23, spin-orbit coupling matrix element (SOCME) values between $S_1$ and $T_1$ for TAHTCNB were calculated to be very low (0.01 cm$^{-1}$) than the other co-crystal systems, further supporting TAHTCNB is a two-state ($^1$CT and $^3$CT) TADF emitter[50]. Furthermore, to reveal the dissimilar intermolecular NCIs that induced the emission from excited states, the functions of reduced density gradient (RDG) and sign ($\lambda_2$)$\rho$ were calculated for all the co-crystals using TDDFT geometry with Multiwfn software[51]. RDG analysis confirmed the presence of clear high density of weak interactions (green region) and more considerable steric hindrance (brown region) between D and A segments in TAHTCNB as compared to TAHOFN, TAHTFPN_G, and TAHTFPN_O, which could effectively restrict the molecular vibrations and prevent energy loss of the excited molecules and revealing TADF susceptibility (Fig. 7a-iii, b-iii, c-iii, d-iii). The combination of cis-conformation of the electron-donating TAH molecular structure and the strong electron-withdrawing benzonitrile moiety are beneficial for the sterically stressed intermolecular-CT characteristics, that collectively leads to small $\Delta E_{ST}$ (0.02 eV) between $S_1$ and $T_1$, and promotes an efficient RISC process that results in TADF emission[52]. Further, to understand the CT nature for all the co-crystals, natural transition orbital (NTO) distributions of lowest singlet and triplet states were calculated from their respective TD-DFT geometry (Included in the Supplementary Figs. 43–46). In particular, the NTOs of the lowest singlet state ($S_1$) for TAHOFN clearly showed a hybrid FMOs distribution, combination of locally excited (LE) and CT in nature. Due to weak acceptor strength of OFN, TAHOFN exhibited weak TSCT. However, TAHTFP_O and TAHTCNB co-crystals exhibited a complete separation of FMOs in the $S_1$, depicting stronger TSCT and therefore they exhibited red-shifted absorption and emission behavior. Therefore, by modulating the different acceptor strength we could easily tune the excited state properties, while geometrical isomerism further assisted in harvesting triplets and exhibiting TADF behavior as illustrated in Fig. 7e. TAHOFN co-crystal showed weak fluorescence, attributed to the close π-π interaction and static emission quenching of TAH via photoinduced electron transfer (PET) mechanism[53] (Supplementary Figs. 47–49). Furthermore, electron spin resonance (ESR) showed a prominent ESR signal (g factor = 2.145 at room temperature) observed for TAHTCNB, which consisted of the ground state TSCT interaction character[12,16]. A significant ESR signal for TAHTCNB originates due to the strong electronic coupling and TSCT interaction with dense segregated stacking, resulting in large red-shifted delayed emission as compared to other mixed stacked weak TSCT complexes (Supplementary Fig. 50). In order to understand the variable CT-emission mechanism accurately, the degree of TSCT ($\rho$) was calculated for all four co-crystals. Correspondingly, single-crystal X-ray geometry displayed the change in C-C bond length of acceptors in CT-complexes, as compared to the neutral form of OFN, TFPN, and TCNB, which resulted in the structural changes leading to the stretching peak being shifted significantly. Further, the degree of CTs ($\rho$) is defined as $\rho$ ($0 \leq \rho \leq 1$), which denotes physical properties like conductivity and

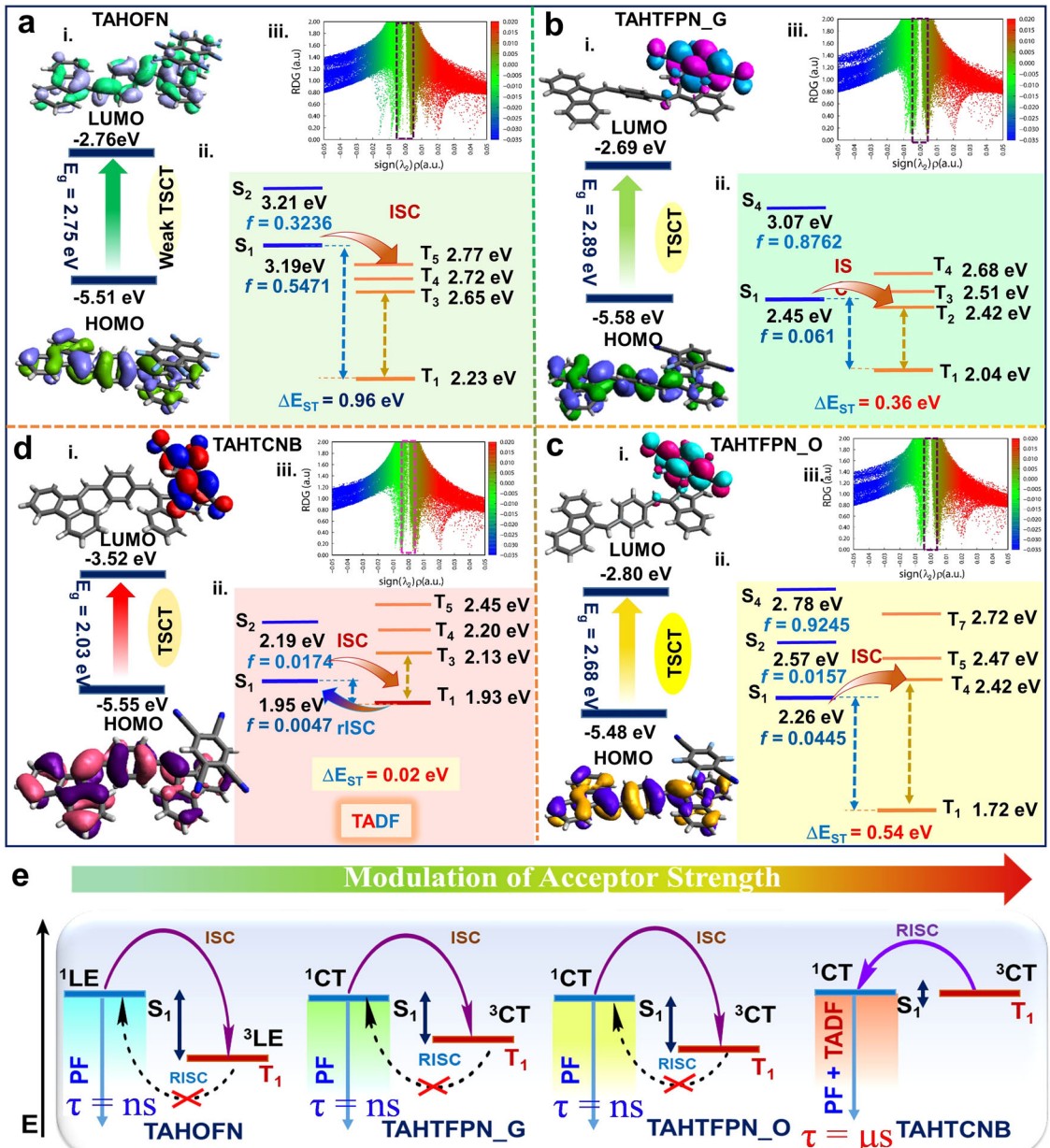

**Fig. 7 | Computational calculations and emission mechanism.** TD-DFT study using Gaussian 16 at B3LYP/631 G (d,p) level. **a**-i, **b**-i, **c**-i, and **d**-i Frontier molecular orbitals (FMOs); HOMO/LUMO distributions and bandgaps. **a**-ii, **b**-ii, **c**-ii, and **d**-ii Emission energy calculations for different singlets/triplets with their corresponding energy splitting ($\Delta E_{ST}$). **a**-iii, **b**-iii, **c**-iii, and **d**-iii The functions of reduced density gradient (RDG) and Sign ($\lambda_2$) $\rho$ iso-surface map with an isovalue of 0.5 for TAHOFN, TAHTFPN_G, TAHTFPN_O, and TAHTCNB respectively. **e** Schematic illustration of excited state modulation at different acceptor strength optical properties of co-crystals.

packing of co-crystals, and can be obtained from Eq.-2[54]. A modest 0.915e, 0.725e, 0.580e, and 0.229e $\rho$ value for TAHOFN, TAHTFPN_G, TAHTFPN_O, and TAHTCNB confirms the gradual decrease of CT degree because of the weakening of the interactions between the pair of D-A molecules[55]. Subsequently, the high value of $\rho$ reveals mixed stack packing mode while relatively low $\rho$ represents segregated stack packing mode[16].

$$\rho = \frac{1}{2}\left(\left[1 - \frac{x_{CT} - y_{CT}}{x_N - y_N}\right] + \left[1 - \frac{z_{CT} - y_{CT}}{z_N - y_N}\right]\right) \quad (2)$$

Where x, y, and z refer the bond lengths assigned in OFN, TFPN, and TCNB, the subscript CT and N denote when the bonds are in CT complex, and the bonds in the neutral state of acceptors (Supplementary Fig. 51).

## Optical waveguide study

The optical wave guiding experiments were performed for individual rod-like micro co-crystals using a confocal microscopy setup in a transmission-mode geometry. A diode laser 405 nm was used as an excitation source, exciting at the left terminus of the microcrystal generating bright greenish luminescence for TAHOFN and TAHTFPN_G, while bright orange-red emission for TAHTCNB microcrystals subsequently propagating to their opposite end. Microcrystals were excited at different points along long axes. The micro-PL images of TAHOFN, TAHTFPN_G, and TAHTCNB show strong greenish and orange-red emissions at the tips of the respective microcrystals (Fig. 8a–c). Strikingly, the PL spectrum showed a narrow bandwidth spectrum covering 450-700 nm with distinct fluorescence and TADF emission peaks centered at 545, 550, and 600 nm for TAHOFN, TAHTFPN_G, and TAHTCNB, respectively (Fig. 8d–f), which matched exactly with the PL

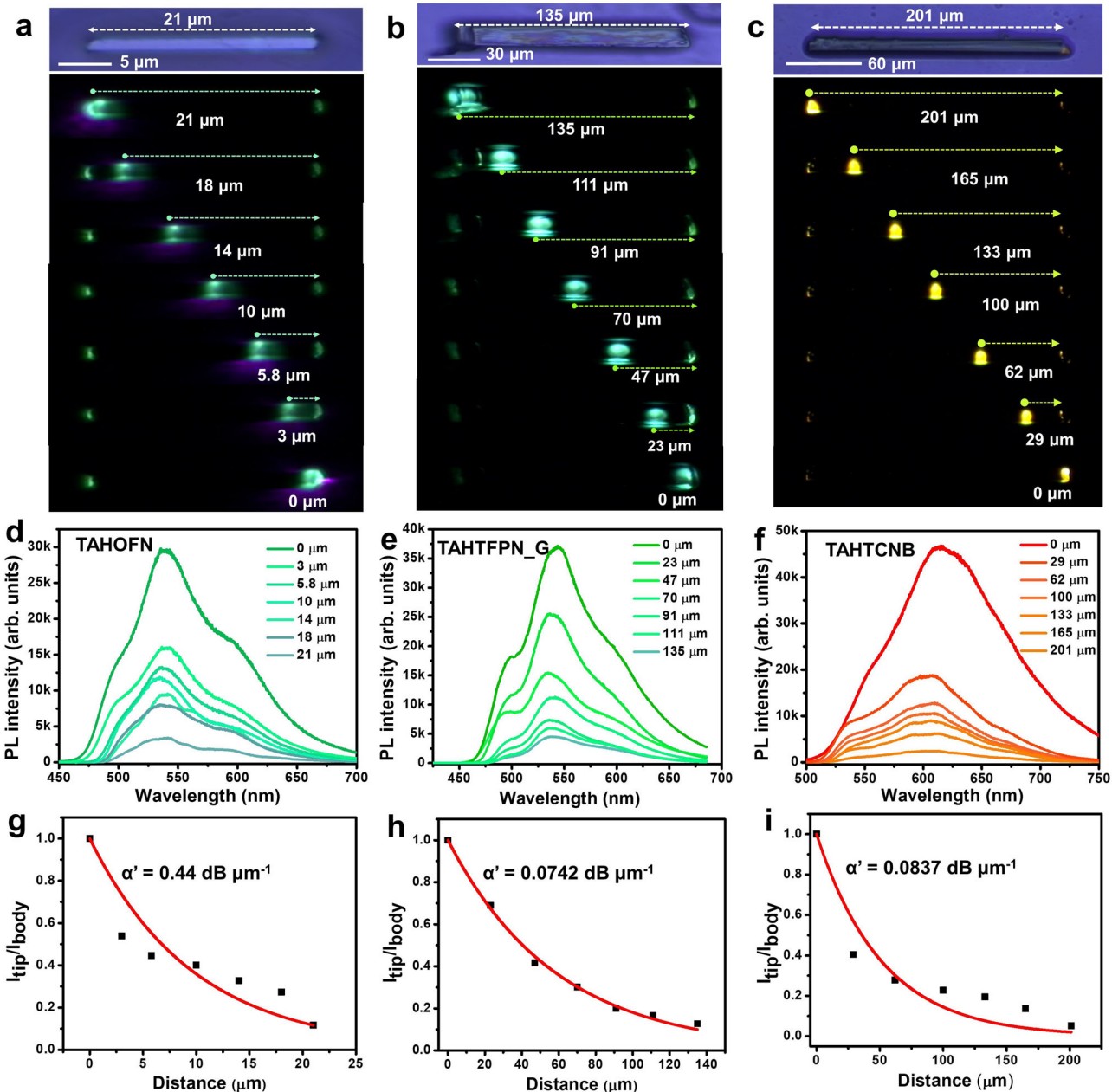

**Fig. 8 | Optical waveguide study. a–c** Micro PL images of TAHOFN, TAHTFPN_G, and TAHTCNB microcrystals; a focused 405 nm excitation laser source was used to excite micro-crystals at seven different positions. **d–f** Emission spectra at the end of the microcrystals for TAHOFN, TAHTFPN_G, and TAHTCNB, the bright line is the emission spectra at the illuminated position. **g–i** Estimation of the optical loss coefficient ($\alpha'$) from the plot of $I_{tip}/I_{body}$ versus the light propagation distance.

emission observed for crystalline powder forms of the co-crystals[56]. Noticeably, the spectral profiles did not shift with the excitation position. However, the PL intensity gradually decreased at the left terminus with the increase of distance from the excitation point to the terminus. Further, spatially resolved spectra for all three microcrystals were recorded by keeping the excitation position constant and PL signals were collected at different positions for TAHOFN, TAHTFPN_G, and TAHTCNB microcrystals (Supplementary Figs. 52–54). The optical loss co-efficient ($\alpha'$) was obtained from distance-dependent PL spectra, whereas the PL intensities were recorded at the illuminated position ($I_{body}$) and collected at the terminus of the crystals ($I_{tip}$) within the micro-rod crystals (Fig. 8g–i). By fitting the exponential decay function, $I_{tip}/I_{body} = \exp(-\alpha'D)$[57], where D is the distance between the illuminated position and the collection position of the crystals, and $\alpha$ is the loss

coefficient (where $\alpha' = 4.34\alpha$), the $\alpha'$ values were determined to be as low as 0.44 dB/µm for TAHOFN, 0.0742 dB/µm for TAHTFPN_G and 0.0837 dB/µm for TAHTCNB, respectively, suggest very good optical waveguide performance of this crystalline organic fluorescent and TADF microcrystal. Notably, these $\alpha'$ values are lower than most of state-of-the-art 1D co-crysytal optical waveguides (Supplementary Table 24)[18]. The observed efficient optical waveguide properties and low $\alpha'$ value for TAHTFPN_G rather than TAHOFN can be explained by a few important factors such as large CT, high PLQY, well-defined crystallinity, defects-free packing, smooth surface and weak coupling between µ and light propagation, leading to no reabsorption and a low loss of the optical waveguides[58]. On the contrary, $\alpha'$s value of TAHCNB is comparatively higher than TAHTFPN_G microcrystal because of triplet contribution in the TADF emission.

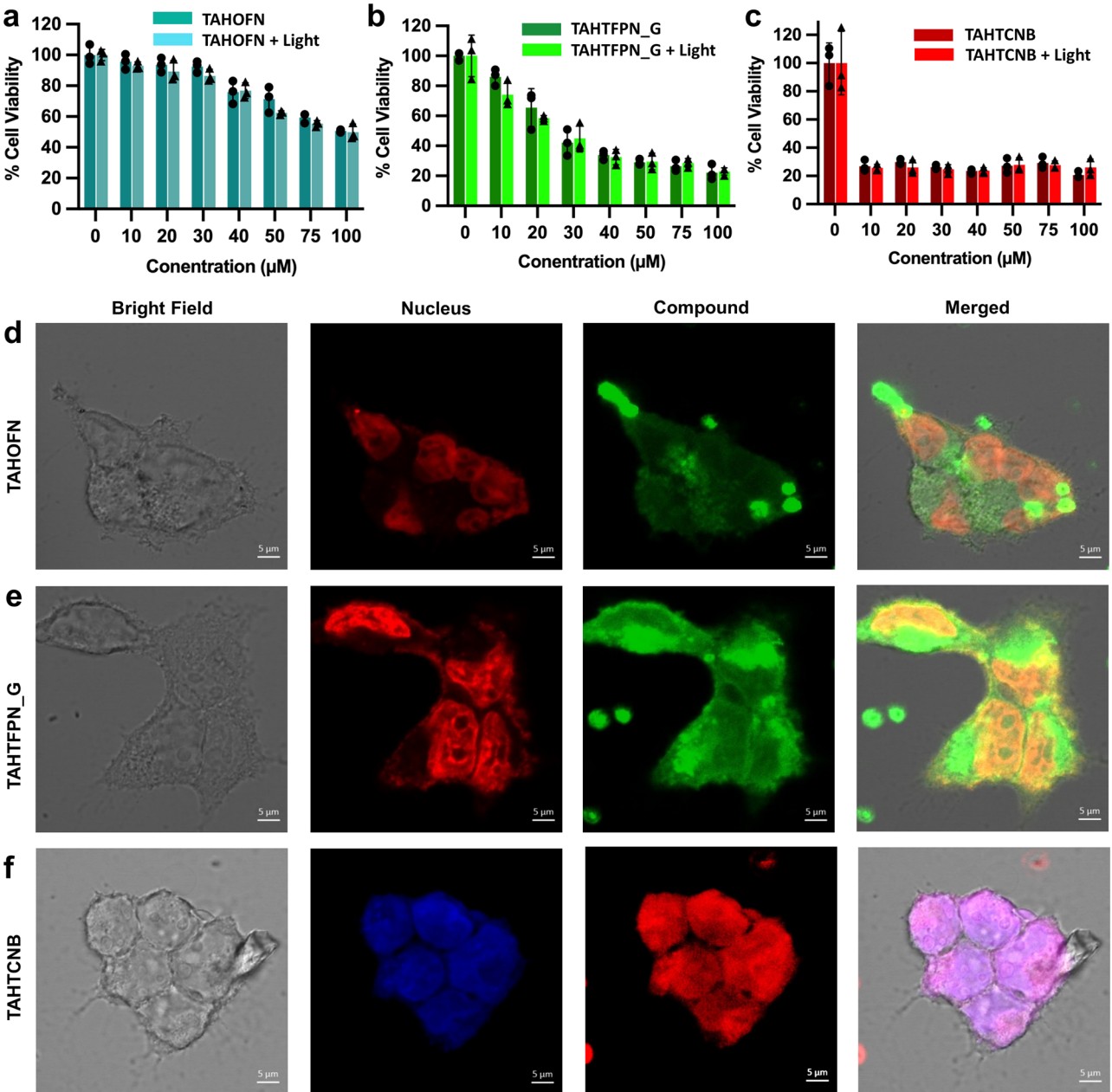

**Fig. 9 | Toxicity and cell imaging study. a–c** Cell viability of different co-crystals-treated MCF-7 cancer cells under dark and white light irradiation (60 mW cm⁻², 5 min). **d–f** Confocal images of MCF-7 cancer cells after incubation them with TAHOFN, TAHTFPN_G, and TAHTCNB co-crystals (10 µg mL⁻¹) for 2 h; the fluorescence of the co-crystals was collected between 500 and 650 nm upon excitation at 488 nm and 561 nm lasers as excitation sources. All the images share the same scale bar of 5 µm. Data are shown as the mean ± SD (*n* = 3). The experimental results shown in figure **a–c**, and **d–f** were independently replicated three times, demonstrating consistent outcomes.

## Cell viability assay and imaging study

The engineered co-crystals exhibited excellent optical properties and dispersibility in water as shown in Fig. 5a–f and Supplementary Figs. 30–33. Therefore, as a rare application of co-crystals, the dark cytotoxicity and the light-induced cell death were quantified using an MTT-based cell viability assay. For this MCF-7 (breast cancer) cells were treated with increasing concentrations of the co-crystals. As shown in Fig. 9a–c, we observed dose-dependent toxicity of TAHOFN and TAHTFPN_G. From the viability calculations, it was found that TAHTFPN was more cytotoxic than TAHOFN. Surprisingly, TAHTCNB caused significant cell death at the lowest concentration, which could be presumably due to the introduction of increasing cyano functional groups in the co-assembly systems. There was no significant difference in the cell viability for all the molecules with or without white light irradiation. Fluorescence microscopy-based uptake study was carried out to confirm the internalization of PSs molecules inside the cells. The MCF-7 cells incubated with co-crystals (at 10 µM) and counterstained with propidium iodide (PI) and 4′,6-diamidino-2-phenylindole (DAPI) for the nucleus of TAHOFN, TAHTFPN_G, and TAHTCNB were imaged using a confocal microscope. The protocol to evaluate the long-term imaging potential of the probes has been schematically presented in Fig. 9d–f and Supplementary Figs. 55–57. The compound panel shown in Fig. 8d–f, corresponds to the fluorescence signal collected from co-crystals. Interestingly, fluorescence in the cytoplasm of the MCF-7 cells confirms the successful internalization of co-crystal molecules inside cells. Additionally, nuclear staining with PI confirms that co-crystal

molecules were localized in the cytoplasm of the cells. This ensures the possibility of imaging application of the fluorescent co-crystals. The mechanism of the internalization of the co-crystals was studied by measuring the fluorescence intensity of the internalized molecules in the presence of endocytosis inhibitors (Supplementary Fig. 58). Uptake of the TAHOFN, TAHTFPN_G, and TAHTCNB decreased significantly, suggesting the active and energy-dependent mechanism of the endocytosis[59].

To inhibit uptake via clathrin and caveolin-mediated pathway, chlorpromazine and beta-methyl cyclodextrin were added to the cells before co-crystal incubation, respectively. Chlorpromazine addition significantly reduced the uptake of TAHOFN (64.9%), and TAHTFPN (75%), while beta-methyl cyclodextrin reduced the internalization of all three co-crystals (TAHOFN = 36.4%, TAHTFPN_G = 43.7% and TAHTCNB = 50.3%). Similar to chlorpromazine, amiloride, and micro-pinocytosis inhibitor reduced the internalization of TAHOFN (55.9%) and TAHTFPN (54.6%). Uptake studies with TAHOFN and TAHTFPN co-crystals confirmed the key role of the clathrin and macropinocytosis-mediated endocytosis, while TAHTCNB was predominantly internalized via caveolin-mediated endocytosis. As the co-crystals were taken up by the cells through the process of endocytosis, they underwent internalization and eventually localized within the cytoplasm. In the case of the co-crystals, this mechanism allowed them to be enclosed within endosomes. The presence of the co-crystals intact within the endosomes signifies their successful internalization by the cells and suggests that they are being processed by the cellular machinery.

## Discussion

In summary, exploring unique photofunctional luminescent co-crystals, mechanistic understanding of color-tunable solid-state emission, and finding appropriate applications is a formidable challenge for a purely organic multicomponent organic system. This work presents a series of rationally designed color-tunable CT co-crystals referred as TAHOFN, TAHTFPN_G, TAHTFPN_O, and TAHTCNB, engineered by utilizing the non-planar twisted donor TAH and planar acceptors OFN, TFPN and TCNB. The co-crystals, TAHTFPN_G and TAHTFPN_O, possess different fluorescence color-specific polymorphs. At the same time, TAHTCNB is an unusual configurational cis-isomer exhibiting TADF feature in a pure hydrocarbon donor-based co-crystal system. Further, a comprehensive agreement of experimental and theoretical results confirmed co-formers' complete involvement and the formation of co-crystals and their structural modulations with intrinsic excited state dynamics for bright and delayed fluorescence. With the increasing acceptor strength, the co-crystals exhibited a decrease in energy band gap and $\Delta E_{ST}$ including variable degrees of through-space CT characteristics leading to different tunable optical properties, blue/red-shifted fluorescence, up-conversion luminescence including singlet and triplet harvesting optical waveguide performances. These, co-crystals exhibited different multiple intermolecular noncovalent bonding, TSCT and π-π interactions which regulated different stacking modes such as H-type mixed stack and J-type segregated stack between co-former units and resulted in dynamic co-crystals with improved physicochemical functions. Consequently, the combined investigations of dense crystal packing, regular supramolecular organization, morphology, the desired orientation of μ, and variable CT in the co-crystals resulted in improved OWGs by self-guiding the generated fluorescence and TADF, through light propagation along its longitudinal axis at ultra-low optical loss co-efficient ($\alpha'$) respectively. These observations are poised to lead the design of luminescent co-crystals in the future for advanced optoelectronic materials development that are of transductive to near-IR frequencies where most of the current growing areas involving fiber-optic communications are conducted. More interestingly, these TAH-based co-crystals exhibited aggregation-induced enhanced emission in water, and their excellent

dispersibility endows good cellular internalization with bright cell imaging performances. Unlike bright cell imaging, the viability study showed that TAHTFPN was more cytotoxic than TAHOFN, while TAHTCNB caused significant cell death at the lowest concentration. Collectively, this work explored unique classes of multifunctional organic co-crystals with very bright color-tunable emission in the solid state as well as enhancing aggregated state emission behavior. These results are precisely described the unusual structure-functionality relationship, achieving high excitons utilization efficiency, low-loss wave-guiding performances. Further exploration in biological applications paved the way for developing organic hydrocarbon-based co-crystals for cellular internalization and potential imaging and therapeutic applications.

## Methods
### Materials
1,4-Bis(fluorenylidenemethyl)benzene denoted as TAH prepared using potassium hydroxide-90% (KOH), 9H-fluorene-98% and terephthalaldehyde-99%. Octafluoronaphthalene-96% (OFN), Tetra-fluoropterepthanitrile-99% (TFPN), and 1,2,4,5-Tetracyanobenzene-97% (TCNB) were purchased from Sigma-Aldrich and used without further purification. Tetrahydrofuran (THF) and Ethanol (EtOH) were purchased from local vendors and further dried and purified.

### Preparation of TAH
In a 100 ml two-neck round-bottom flask, a mixture of potassium hydroxide (1 g, 18 mmol) and 9H-fluorene (3 g, 18 mmol) was dissolved in absolute ethanol and refluxed for 1 hour under Argon gas. 2 g of benzene-1,4-dicarbaldehyde (15 mmol) was added into the reaction mixture and refluxed for 12 hours. The yellow color precipitate was filtered and washed several times using ethanol to give pure product as a bright yellow solid powder compound TAH. (Yield: 75 %); [1]H NMR (400 MHz, Chloroform-*d*) δ(ppm): 7.83 (d, J = 7.0 Hz, 1H), 7.75 (d, J = 6.4 Hz, 4H), 7.71 (s, 2H), 7.44-7.39 (m, 1H), 7.36 (dd, J = 11.5, 7.9 Hz, 2H), 7.16-7.10 (m, 1H).[13]C NMR (100 MHz, Chloroform-*d*) δ(ppm): 141.42, 139.51, 139.24, 136.96, 136.74, 136.53, 129.59, 128.73, 128.35, 127.08, 126.81, 126.73, 124.46, 120.31, 119.85, 119.66.

### Preparation of co-crystals
TAHOFN co-crystal was prepared by using 1:1 stoichiometric mixture of TAH (500 mg, 1.17 mmol) and OFN (316 mg, 1.17 mmol) dissolved in anhydrous THF and kept at RT. When, THF was evaporated a yellowish color solution turns to rod-like green crystals after 1 day.

TAHTFPN_G co-crystal was prepared by using 1:1 stoichiometric mixture of TAH (500 mg, 1.17 mmol) and TFPN (232 mg, 1.17 mmol) dissolved in anhydrous THF and kept at RT. When, THF was evaporated a yellowish color solution turns to rod-like green color crystals after 2 day.

TAHTFPN_O co-crystal was prepared by using 1:1 stoichiometric mixture of TAH (500 mg, 1.17 mmol) and TFPN (232 mg, 1.17 mmol) dissolved in anhydrous THF and 60 °C temperature applied. When, THF was evaporated a yellowish color solution turns to rod-like orange color crystals were obtained after 2 day.

TAHTCNB co-crystal was prepared by using 1:1 stoichiometric mixture of TAH (500 mg, 1.17 mmol) and TFPN (207 mg, 1.17 mmol) dissolved in anhydrous THF and kept at RT. When, THF was evaporated a yellowish color solution turns to rod-like red color crystals obtained after 2 day.

### Measurements
**FTIR and Raman spectral analysis.** Fourier transform infrared (FTIR) spectra of all samples were recorded in a Perkin Elmer instrument in attenuated total reflectance (ATR) mode. Raman spectra analysis was performed using a laser micro-Raman system (Horiba Jobin Vyon, LabRam HR) with 633 nm laser excitation.

**X-ray diffraction analysis.** Rigaku SmartLab X-ray diffractometer equipped with copper Kα ($\lambda = 1.54$ Å) as the source with 9 kW power was used for the measurements of powder X-ray diffraction in the powder and thin films state. The XRD patterns were measured for the angle of $2\theta$ range of $5° - 50°$ at the scan rate 0.3°/s. Whereas, Single-crystal structures (SCXRD) of TAHOFN, TAHTFPN_G, TAHTFPN_O, and TAHTCNB were obtained using Agilent X-ray diffractometer. The single-crystal x-ray diffractometer is equipped with Mo X-ray source (Mova), CCD detector (Eos), Oxford cryo-system (80-500 K) and crystal AlisPRO software and Autochem softwares. All the data were collected at 293 K. All co-crystal structures were solved by SHELXT using direct methods. All the non-hydrogen atoms were located by SHELXT directly and refined anisotropically by SHELXL2018 with least squares methods. The revised data have been deposited with the Cambridge Crystallographic Data Center (CCDC) (accession codes CCDC 2109333, CCDC 2109336, CCDC 2109335, and CCDC 2109337). The stacking structure and intermolecular interactions of the co-assembly were analyzed using Mercury and Materials Studio software.

**Solid-state absorbance and emission studies.** The absorption and photoluminescence spectra were measured using PerkinElmer, Model Lambda-25 spectrometer and Horiba-Fluoromax4 with Varian Cary Eclipse spectrometer, respectively. The absolute photoluminescence quantum yields (PLQY) were measured using the Horiba Fluoromax (Jobin Yvon equipped with Integrating sphere) absolute quantum yield spectrometer. The time-resolved photoluminescence lifetimes (TRPL) were investigated using an Edinburgh Life Spec II instrument. Temperature-dependent (100–300 K) photoluminescence (PL) and delayed fluorescence lifetime measurements were carried out using a liquid nitrogen-cooled optical cryostat (Optistat, Oxford Instruments) attached to an Edinburgh FSP-920 instrument.

**Transient absorption and laser-induced fluorescence measurements.** For TA and LIF spectral data acquisition LP980-KS spectrometer with a visible PMT detector and an ICCD detector was used. These detectors were mounted on different ports of the LP980 spectrograph, allowing rapid switching between spectral and kinetic detection. For the low temperature measurements an Oxford Instruments OptistatDN cryostat was used in the sample chamber of the LP980 and temperature was controlled by operating software, L900. At the room temperature studies were done using the slides were either mounted inside the cryostat or in a film sample holder (L-F04). Pump pulses at 355 nm were produced by a Litron Nano S 130-10 laser. The pump pulse width was ~5 ns and its energy were ~20 mJ/pulse. A Xe lamp in the LP980 generated the white-light probe beam, operating in pulsed mode and producing pulses with a duration of 6 ms. Pump only and probe only backgrounds were subtracted automatically by the L900 software when necessary. Besides, for the LIF measurements a 370 nm long-pass filter was used before the emission monochromator to eliminate the interference from scattered pump light on the spectra.

**EPR study.** Electron Paramagnetic Resonance (EPR) spectra for the co-crystal's powders were recorded on a JES-FA200 ESR spectrometer, at room temperature with microwave power, 0.995 mW; microwave frequency, 9443.961 MHz; and modulation amplitude, 2.

**Microscopic studies.** The surface morphology of the co-crystals was performed using scanning electron microscopy (FESEM) (Zeiss, Model: Sigma-300). All drop casted samples were gold-sputtered under vacuum to achieve a thin layer of conductive gold coating on the micro/nano crystalline samples. While, atomic force microscopy (AFM) analysis were performed using drop-casted sample over silicon substrate, in AFM Asylum Cypher, Oxford Instruments. For more clear morphology and crystallinity information of the co-crystals Transmission electron microscopy (TEM) images, and selected area electron diffraction (SAED) patterns were recorded in the films. The samples were prepared using 5 μL of the sample solution was drop-casted on carbon coated copper grid (300 mesh Cu grid with thick carbon film from Pacific Grid Tech, USA) and allowed to air dry for 2 minutes and then the excess sample was soaked up with a tissue paper. The grid was then immediately freeze-dried and the FETEM images were taken in JEOL JEM-2100F microscopes.

**Cyclic voltammetry study.** Cyclic voltammetry (CV) studies were carried out for electrochemical measurements using CH Instruments 760D electrochemical workstation at a scan rate of 50 mV/s. A three-electrode cell was employed having platinum wire as a counter electrode, glassy carbon as a working electrode and Ag/Ag+ for reference electrode. Tetrabutylammonium hexafluorophosphate (0.1 M) in acetonitrile solvent was used as supporting electrolyte (Fc+ /Fc couple was used as internal reference). A drop-casted thin film on the working electrode using 1 mM solution of each sample dissolved in THF was used for the measurements performed at room temperature under inert atmosphere.

**Thermal properties measurements.** The melting transition temperatures and associated enthalpy changes for all the co-crystals were determined on a differential scanning calorimeter (Q20 DSC) under inert atmosphere at a heating rate of 10 °C/min. To investigate the thermal stability of the co-crystals, thermogravimetric analyses (TGA) were studied by a Mettler-Toledo TGA/SDTA 851e thermogravimetric analyzer in a temperature range of 30–600 °C under nitrogen atmosphere at 5 °C min$^{-1}$ heating rate.

**Computational Studies.** To investigate the highest occupied molecular orbital (HOMO), lowest unoccupied molecular orbital (LUMO) and energy levels at different spin multiplicities, density functional theory (DFT) and time-dependent density functional theory (TD-DFT) calculations were performed by employing the combination of Becke3-Lee-Yang-Parr (B3LYP) hybrid functional and 6-31 G (d,p) basis set using the Gaussian 16 package. Natural transition orbitals (NTOs) for each lowest excited state of the co-crystals was calculated using Multiwfn. Further, spin-orbit coupling (SOC) studies were performed using Orca 4.2.0 software package.

The simulated growth morphologies of co-crystals were calculated by using the Materials Studio software, based on the attachment energy theory. The molecular structure was first optimized on the basis of the experimental crystal structure. The geometric and energy calculations were performed using the Forcite and Morphology modules.

Combined with the SCXRD crystal data, the Hirshfeld surfaces, electrostatic potential, and molecular orbital were simulated by CrystalExplorer 3.1.51 The Hirshfeld surfaces and the electrostatic potential (ESP) were mapped with dnorm using STO-3G basis set at the Hartree-Fock theory.

**Confocal micro spectroscopy studies.** The Confocal Micro Spectroscopy images were carried out using a Wi-Tec alpha 300R laser confocal optical microscope facility equipped with a Peltier-cooled CCD detector. 405 nm laser was used as an excitation source. The excitation and collection of signals from the output of the microstructure were performed by an upright microscope (×20; NA: 0.6). The output signal collection was performed using ×20 objective for every 0.3 ms and the signal was sent to a CCD detector through a multimode optical fiber of diameter 100 μm (core). The time taken to complete one set of experiment is dependent upon the solvent evaporation time. All measurements were performed at ambient condition and images were processed by using WI-TEC 2.0 software.

All other fluorescence microscopic images were captured using a ZEISS Axio Vert.A1 inverted microscope with ×10 objective. All digital pictures were taken with a canon power shot SX420 IS digital camera.

**Method section for cells.** MCF-7 breast cancer cells (Cat.no. BR-MCF7-MEM-NEAA) were procured from the National Centre for Cell Science in Pune, India. The cells were cultured in a $CO_2$ incubator using DMEM (Dulbecco's Modified Eagle Medium) supplemented with 10% fetal bovine serum and a 1% penicillin and streptomycin antibiotic solution. The authenticity of the MCF-7 cells was confirmed through STR phenotyping, and a test conducted on 04/07/2023 verified the absence of mycoplasma contamination.

**Cell viability assay.** The effect of TAHOFN, TAHTFPN_G, and TAHTCNB on cell viability was studied on human, MCF7 (breast cancer cells). Firstly, cells were seeded on 96 well plates at the density of 6000 cells per well. Plates were incubated in $CO_2$ incubator overnight for cell attachment. Following this, cells were treated with TAHOFN, TAHTFPN_G, and TAHTCNB at various concentrations. After 24 hours of treatment, MTT dye was added to individual wells (5 μL, 5 mg/mL in PBS), and plates were incubated for 2 h. In the end, to measure the absorbance from MTT dye, 150 μL DMSO solution was added to each well, and absorbance was recorded at 470 nm in the multiplate reader (Infinite 200 PRO, TECAN). Compusyn software was used for combination index analysis.

**Long-term cellular imaging.** To evaluate the imaging efficiency of TAHOFN, TAHTFPN_G, and TAHTCNB inside the cellular environment, MCF-7 cells were seeded on 96-well plates at a density of 6000 cells per well. TAHOFN, TAHTFPN_G, and TAHTCNB were dissolved in DMSO at a concentration of 10 mM. The co-crystals were then dissolved in DMEM medium to achieve a concentration of 10 μM for addition to the cells. After a 6-hour incubation, the cells were washed with PBS three times and fixed with 4% formaldehyde. Images were captured using a confocal microscope with 488 nm and 561 nm lasers as excitation sources.

**Internalization mechanism study.** To study the mechanism of the internalization of the co-crystals, MCF7 cells were seeded in six-well plates at the density of 200 K cells per well. Cells were treated with co-crystal at 10 μM concentration for 1 h. Following this, cells were washed, trypsinized and the fluorescence intensity of the cells was measured with TECAN 96 well plate reader.

**Statistics and Reproducibility.** In Figs. 4a–l; 6d–f; 8a–c; and 9d–f, each experiment was repeated independently three to four times with consistence results.

**Reporting summary**
Further information on research design is available in the Nature Portfolio Reporting Summary linked to this article.

## Data availability
Supplementary information file contains all additional experimental data, tables, and results. The data that support the findings of this study are available from the authors on request. The X-ray crystallographic coordinates for structure have been deposited with the Cambridge Crystallographic Data Center (CCDC) (accession codes CCDC 2109333, CCDC 2109336, CCDC 2109335, and CCDC 2109337). These data can be obtained free of charge from The Cambridge Crystallographic Data Center via www.ccdc.cam.ac.uk/data_request/cif. The source data underlying main manuscript Figs. 3–9 and Supplementary Figs. 2–58 are provided as a Source Data file. Source data are provided with this paper.

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

## Acknowledgements

Financial grants from Department of Science and Technology, New Delhi, India project no. DST/CRG/2019/002614, Deity, India No. 5(1)/2022-NANO, ICMR, Grant no. 5/3/8/20/2019-ITR and Max-Planck-Gesellschaft IGSTC/MPG/PG(PKI)/2011A/48 are acknowledged. Central Instruments Facility, Department of Chemistry, and Centre for Nano-technology, IIT Guwahati are acknowledged for providing various instrument facilities. R.C. thanks SERB-New Delhi (CRG-2018/001551)

and UoH-IoE for financial support and also the School of Chemistry and Centre for Nanotechnology for instrument facilities. D.B. gratefully acknowledge the MHRD, Govt. of India for providing the necessary funding and fellowship to pursue this research work.

## Author contributions

Deba.B. designed and synthesized co-crystals, performed structural and photophysical characterization, analysis, computational calculations, and prepared the first draft of the manuscript. M.A. and R.C. performed the optical waveguide study, analysis and were involved in manuscript preparation. A.P.B. and S.S.G. performed biological studies. P.R. executed optical studies, analysis and involved in manuscript preparation. Debi.B. performed NMR and optical studies. P.K.I. supervised the overall project, and was involved in planning, analysis, and manuscript preparation.

## Competing interests

The authors declare no competing interests.
