## [Peer Review File · Nature Communications]

Highly Efficient Color-Tunable Organic Co-crystals: Unveiling Polymorphism, Isomerism, Delayed Fluorescence for Optical Waveguides and Cell-imagingREVIEWER COMMENTS

Reviewer #1 (Remarks to the Author):

Molecule-based cocrystals, composed of two or more different molecules, provide the ability to regulate many physiochemical properties, such as solubility, stability, and photonic/electronic performances by designing and controlling the crystal stacking modes and noncovalent intermolecular interactions. In this work, Iyer et al. designed and synthesized four twisted aromatic hydrocarbons (TAHs) luminescent cocrystals (i.e., TAHOFN, TAHTFPN_G, TAHTFPN_O, and TAHTCNB) via solvent diffusion method. Combined experimental and theoretical analyses suggested that the prepared cocrystals showed tunable energy band gap, variable degree of charge transfer and multiple noncovalent intermolecular interactions between the aromatic cores. Due to the defined packing with directionality and specificity, these cocrystal have highly luminescent efficiency (~ 77%) and superior waveguide performance. Besides, these cocrystals exhibited unusual aggregation induced emission (AIE) with water dispersibility, endowing good cell imaging performances. In all, this work reported a new class of multifunctional organic cocrystals with highly color-tunable emission, which might pave the way for developing organic hydrocarbon based cocrystals and their potential applications. I suggest that this work can be potentially published in Nature Communications upon further revision. Detailed comments and suggestions are listed below:

1. Why is the H-type mixed stacking mode formed in TAHOFN and TAHTFPN_G, whereas the J-type segregated stacking mode formed in TAHTFPN_O and TAHTCNB? Could authors provide the principles of materials design?
2. The experimental PXRD data of the cocrystals are not exactly matched with the simulated data, which means the cocrystals may not be the pure phase. Could authors explain this situation?
3. In Fig. 2d, cocrystal TAHTCNB starts to decompose around 100 °C, which is less stable compared to the other three cocrystals. However, authors claimed that TGA study reveals the formation of cocrystals and relatively high thermal stability of TAHTCNB (235 °C) rather than the other three cocrystals. Please explain.
4. Figure 4i: why the lifetimes of the samples exhibited obvious different decay?
5. For the lifetime results from the decay curves in Fig. 4i and Fig. 4l, the fitting results and fitting goodness should be provided.
6. Polymorphs and cocrystals as luminescent materials and wide applications are hot topics, to arouse broader readerships, some strongly related works on luminescent molecular polymorphs and cocrystals (Angew. Chem. Int. Ed. 2011, 50, 12483; ; ACS Appl. Mater. Interfaces, 2018, 10, 22703; Angew. Chem. Int. Ed. 2023, 62, e202217054) and AIE type cocrystals (ACS Central Science, 2020, 6, 1169; Mater. Horiz. 2014, 1, 46) could be compared and included as references.
7. Currently, authors conducted crystal structure analysis after describing the optical properties and mechanisms. I suggest authors first do the crystal structure analysis and then investigate the optical properties and mechanism of cocrystals, which may help readers better understand the relationship between structure and property.
8. For the optical waveguide, the loss efficiency needs to be compared with other organic or hybrid systems in literatures (Sci. Bull. 2022, 67, 2076; Sci. China Chem. 2022, 65, 408).

9. The image resolution needs to be further improved.

Reviewer #2 (Remarks to the Author):

[Note from the editor - please also see attached PDF]

This manuscript reported a series of rationally designed color-tunable CT co-crystals developed by utilizing the non-planar twisted donor TAH and planar acceptors OFN, TFPN and TCNB. The cis-isomeric co-crystal exhibits TADF characteristics, and the applications in optical waveguide and cell imaging have been demonstrated. The overall study is very interesting and attractive. The manuscript is also well organized. Therefore, the manuscript is enough to meet the requirements for publication after addressing the following issue.

1. For TADF mechanism, except for temperature-dependent emission decay curves, the temperature-dependent emission spectra should also be provided. Additionally, it is crucial to fit the delay curves, and we need to know the changes in the proportion of long and short lifetimes at different temperatures to determine whether it is TADF emission.
2. The four microcrystals have different full width at half maximum (FWHM), what is this related to?
3. I cannot understand why co-crystals of TAHTFPN_G and TAHTCNB have higher cytotoxicity. More experimental data should support the author's claim that cyanide is increased in the co-crystals. Because when using co-crystal to stain cells, co-crystal will decompose? What is the mechanism by which co-crystals can be internalized into cells? Which organelle in the cell is being stained? And what are the fluorescent dots outside the cells in Figure 8e-f?
4. The abbreviations that appear for the first time in the manuscript need to be explained, such as PXRD, TGA, DSC, etc.
5. "What does 'UV OFF' mean, does it refer to afterglow emission or under daylight, please check carefully."

Reviewer #3 (Remarks to the Author):

Recommendation: Rejected.

Comments:

This article by Parameswar Krishnan Iyer and co-workers reports a series of organic co-crystals using twisted aromatic hydrocarbon donor and three diverse planar acceptors. They report color-tunable emission in aggregated state via variable packing and through-space charge-transfer interactions. By adjusting the strength of acceptors, structural transformation into hybrid stacking modes generates color-specific polymorphs, an unprecedented configurational cis-isomer with very high photoluminescence quantum yield. Moreover, the cis-isomeric co-crystal exhibits triplet-harvesting

thermally activated delayed fluorescence characteristics, and the 1D co-crystal exhibits efficient photon-transducing optical waveguides. Lastly, bright cell imaging performances were also obtained. However, color-tunable emission in aggregated state, color-specific polymorphs, efficient photon-transducing optical waveguides and bright cell imaging performances have ever been reported for other organic cocrystals. In a word, the novelty of this work is not enough, many data and expressions are not accurately. It is recommended to submit to other journals.

Here are few specific comments:

1. There are lots of Grammatical errors :

a) "Remarkably, by adjusting the strength of acceptors, structural transformation into hybrid stacking modes generates color-specific polymorphs, an unprecedented configurational cis-isomer with very high photoluminescence quantum yield." "To overcome this issues twisted aromatic hydrocarbons (TAHs) can play a crucial role,..." "As expected TA spectra depicted a positive signal near 600 nm while life-time was found to be higher than that of 77K (Fig. 4l). In addition PL spectra and TRPL decay profile of TAHTCNB co-crystal under inert/air and variable temperatures", in which connectives (i.e., and) and comma should be added in suitable locations.

b) There are many other kinds of ungrammatical sentences, for example, "Hence, rational design of TAH-based multifunctional co-crystals with improved luminescence in aggregated/solid states, promising photon harvesting characteristics is of great significance which remains an imperative research area so far", "Co-crystals with uniform chain-like molecular arrangements through continuous π - π interactions, their smooth surface and highly rigid crystalline structures, benefited to rationalize the unidirectional photon propagation at very low optical loss coefficient.", and etc.

c) Lots of words were misspelled. For example, the word of col-or-tunable in "...with their respective tunable packing oriented col-or-tunable polymorphs and configurational isomeric binary assembly structures", transductive in "...that are transductive to near-IR frequencies where most of the current growing areas involving fiber-optic-based communications are conducted", co-crystalization in "color-specific polymorphic co-crystals were obtained by controlling the temperature during co-crystalization such as at room temperature (RT)...", color-tunability in "aggregation induced emission, color-tunability and self-guiding singlet-triplet optical waveguides and biological application for the very first time.", co-crystallization in "... investigate more critically the dissimilarity in CT/packing interactions and mechanistically determine the process of co-crystallization.", Estimation in "Estimation of the optical loss coefficient (α) from the plot of I_{tip}/I_{body} versus the light propagation distance.", and so on.

Please check carefully throughout the manuscript.

2. In page 3, line 88, "All the integrated CT-complexes displayed different luminescence behaviour in the presence of different strength of acceptors as shown in Scheme 1.", while only the CT mechanism for TAHTFPN_G and TAHTFPN_O were provided, and the CT character of TAHOFN was not proved.

3: In page 4, line 109, "Besides, TAH co-crystals exhibited unusual AIEE phenomenon with superior water dispersibility, endowing good cellular internalization with bright cell imaging performances." It is very difficult to understand that the co-crystals without any hydrophilic functionalization can present superior water dispersibility.

4. In page 5, line 120, "Four color-tunable fluorescent rod-shaped single-crystalline co-crystals denoted as TAHOFN, TAHTFPN_G, TAHTFPN_O and TAHTCNB were prepared via solvent diffusion method, by using 1:1 stoichiometry ratio of the precursor molecules in THF solvent.", in which the solvent diffusion

method is not correct, because the authors adopted evaporation method to prepare all the co-crystals.

5. In page 5, line 125, "TAH co-assembled very promptly (within 48 hours) and resulted in rod-like four distinct colored CT complexes." It should be better to express clearly which acceptor did TAH co-assemble with.

6. In page 5, line 126 to 128, "Notably, color-specific polymorphic co-crystals were obtained by controlling the temperature during co-crystallization such as at room temperature (RT) for TAHTFPN_G and at 60°C for TAHTFPN_O, while others were grown at RT (Fig. 1a)." Is 60°C the temperature during dissolving? Why do the author only provided the case of TAHTFPN at 60°C? How about TAHOFN and TAHTCNB?

7. In the SI file, "Supplementary Fig. 1. (a) Fluorescence microscopy images for TAH, TAHOFN, TAHTFPN_G, TAHTFPN_O and TAHTCNB co-crystals with the excitation wavelength 405 nm. (b) While at the bottom images are the bright field images.", the caption includes TAH, but the data of TAH cannot be found.

8. In page 7, Fig. 2e., ¹H NMR data collected using CDCl₃ as the solvent. Actually, cocrystals are not stable in organic solvents, which can be well dissolved and the corresponding donor and acceptor exist in isolated single molecular state. Therefore, using liquid ¹H NMR to characterize the cocrystals is not a correct tool.

9. In page 9, Fig. 3., the TEM images of the cocrystals are presented, but the detailed preparation procedures of the TEM samples cannot be found. It should be better to provide it.

10. In page 9, Fig. 3e., the TEM images of TAHOFN give a helical ribbon-like morphology rather than rod, which is not consistent with the results of the SEM data and AFM data.

11. In page 10, line 234, "To investigate the luminescent features of the afforded co-crystals, UV-visible and photoluminescence (PL) studies were performed in solid crystals and aggregated state respectively." Actually, solid crystals are also aggregated state, so the "solid crystals and aggregated state" here can be replaced by "bulk crystals and micro/nano-scale crystals".

12. In page 10, line 236 to 249, "TAHOFN, TAHTFPN_G exhibited blue-shifted absorption at 460 and 430 nm whereas TAHTFPN_O and TAHTCNB showed a red-shifted absorption at 490 and 500 nm related to the absorption of parent TAH (485 nm), deciphering TSCT interactions between constituent precursors." If TAHOFN with a blue-shifted absorption really includes the TSCT interaction? The author only provided the corresponding data about TSCT interaction for the other three cocrystals, i.e., "FMOs distribution suggested that HOMO of the three co-assemblies were mainly located on the donor TAH, whereas LUMO aligned with the acceptors TFPN and TCNB, leading to well separated HOMO and LUMO, clearly indicating the TSCT characteristics of the co-assemblies (Supplementary Fig. 32-33)³⁸."

13. In page 10, line 238, the full name of TSCT should be provided here because of its first occurrence.

14. In page 10, line 247 and 248, "Therefore, the TAH based co-assembly approach reduces aggregation caused quenching effect and showed improved condensed state emission^{10,33}." There is no data that prove ACQ effect of TAH. It should be better to supplement the corresponding data of TAH solution and pure TAH solids, including the FM images, quantum yield, life time, and etc.

15. In page 14, in the caption of Fig. 5., "(a_i, b_i, c_i and d_i)", "(a_{ii}, b_{ii}, c_{ii} and d_{ii})" and "(a_{iii}, b_{iii}, c_{iii} and d_{iii})" should be corrected into "(a-i, b-i, c-i and d-i)", "(a-ii, b-ii, c-ii and d-ii)" and "(a-iii, b-iii, c-iii and d-iii)".

16. In page 15, line 348-349, the expression of "In order to understand the red-shifted CT-emission

mechanism for all four co-crystals accurately, the degree of TSCT (ρ) was calculated" is not correct because only two cocrystals presented red-shifted CT-emission.

17. In page 20, line 469-470, the authors describe "As shown in Fig. 8a-c, we did not observe significant dark toxicity of TAHOFN, even up to 100 μ M concentration." I don't think so, as shown in Fig. 8a, the cell viability is already lower than 80% even under the concentration of 30 μ M, indicating high dark toxicity, which is also proved by Fig. 8d-f.

18. In page 21, line 479, "Microscopy-based uptake study was carried out to confirm the internalization of PSs molecules inside the cells." The full name should be provided here.

19. In page 21, in Fig. 8d-f, because all the nucleus under the three cases were stained with PI, they should present the similar color. Why do the case of TAHTCNB is blue and different from the other two cases?

20. In page 25, line 630, there is an extra "of" in the sentence of "Following the attachment, cells were treated with 10 μ m of of TAHOFN, TAHTFPN_G and TAHTCNB one time and incubated for 6 h."

21. As we all know, the materials applied in cell imaging should be nanoscale, so how did the authors prepare the samples used in cell imaging should be detailed.

Response to Reviewer's Comments for Manuscript ID: NCOMMS-23-09657

Reviewer(s)' Comments to Author:

Reviewer #1 (Remarks to the Author):

Molecule-based cocrystals, composed of two or more different molecules, provide the ability to regulate many physiochemical properties, such as solubility, stability, and photonic/electronic performances by designing and controlling the crystal stacking modes and noncovalent intermolecular interactions. In this work, Iyer et al. designed and synthesized four twisted aromatic hydrocarbons (TAHs) luminescent cocrystals (i.e., TAHOFN, TAHTFPN_G, TAHTFPN_O, and TAHTCNB) via solvent diffusion method. Combined experimental and theoretical analyses suggested that the prepared cocrystals showed tunable energy band gap, variable degree of charge transfer and multiple noncovalent intermolecular interactions between the aromatic cores. Due to the defined packing with directionality and specificity, these cocrystal have highly luminescent efficiency (~ 77%) and superior waveguide performance. Besides, these cocrystals exhibited unusual aggregation induced emission (AIE) with water dispersibility, endowing good cell imaging performances. In all, this work reported a new class of multifunctional organic cocrystals with highly color-tunable emission, which might pave the way for developing organic hydrocarbon based cocrystals and their potential applications. I suggest that this work can be potentially published in Nature Communications upon further revision. Detailed comments and suggestions are listed below:

Author Response: *We thank the esteemed reviewer-1 for carefully reading, reviewing our manuscript and describing its fundamental importance and the very promising results. Also, we are very thankful for providing encouraging feedback and recommendation for the publication of this manuscript in Nature Communications.*

Comment 1. Why is the H-type mixed stacking mode formed in TAHOFN and TAHTFPN_G, whereas the J-type segregated stacking mode formed in TAHTFPN_O and TAHTCNB? Could authors provide the principles of materials design?

Author Response: *We thank the esteemed reviewer-1 for these insightful comments.*

The H-type mixed stacking mode were formed in TAHOFN and TAHTFPN_G, whereas in TAHTFPN_O and TAHTCNB, the J-type segregated stacking mode were formed, which are ascribed by the following aspects.

- i) Through-space charge transfer (CT) complexes are the combinations of charge-donating (D) and charge-accepting (A) organic small molecules, formulated by modulating interactive CT-strength at different crystallization environments. In principle, versatile secondary noncovalent intra/inter-molecular interactions such as hydrogen bonds (HBs), halogen bonds (XBs), π - π stacking including CT played a pivotal role to define directional interactions, and the ordered structures resulted in different hybrid molecular stacking and definite aggregation patterns in multicomponent systems. Particularly, π - π interactions and CT-strength primarily regulate the hybrid stacking mode in co-assembly systems (Chem. Mater. **2016**, 28, 3–16).*

- ii) *Herein, diverse π -acceptor aromatic cores were selected viz. in OFN, the density of aromatic π -clouds at naphthalene core is higher than phenyl cores in TFPN and TCNB. Besides, the strong electron-withdrawing behaviour of $-\text{CN}$ group in TCNB is diffusing the phenyl π -clouds to $-\text{CN}$ rather than poor electron withdrawing $-\text{F}$, therefore they are less likely available for the strong orthogonal π - π stacking. Therefore, a face to face π - π stacking becomes possible between TAH and OFN to form mixed stack as confirmed by their SC-crystal structures. Likewise, TAH and TFPN favors some extent of orthogonal π - π stacking at RT, however, during higher temperature crystallization, the density of π -clouds in phenyl is less available and formed segregated stack TAHTFPN_O as a similar stacking fashion to TAHTCNB.*
- iii) *In addition, computational studies (FMOs) revealed that all four co-crystals exhibited variable through-space CT character owing to the presence of different strengths of acceptors in the order of $\text{OFN} < \text{TFPN} < \text{TCNB}$. From the crystal structures the degree of CT (ρ) and counter pitch angle of a D-A pair was estimated to be 0.915e, 0.725e, 0.580e, 0.229e and 88.7°, 80.10°, 76.91°, 76.03° value for TAHOFN, TAHTFPN_G, TAHTFPN_O and TAHTCNB, respectively. Subsequently, the high value of ρ and counter pitch angle reveals mixed stack packing mode while relatively low ρ and counter pitch angle represent segregated stack packing mode, respectively (Adv. Mater. 2022, 34, 2107169).*
- iv) *Therefore, these dynamic assembly and weak secondary interactions of multi-component co-crystals potentially affords a synergistic influence on the alteration of molecular packing and hybrid stacking modes, which thus further leads to color-tunable PL emission.*

Comment 2. The experimental PXRD data of the cocrystals are not exactly matched with the simulated data, which means the cocrystals may not be the pure phase. Could authors explain this situation?

Author Response: *We thank the esteemed reviewer-1 for seeking this explanation.*

In general, an exactly matching experimental PXRD patterns with simulated patterns indicates that the bulk powder samples of these compounds pack the same as in the single crystals which is rarely found in the literatures. In single crystal XRD, single, discrete diffraction peaks are observed, and crystal structure information is more straightforward to interpret; the spatial properties of a single crystal may not necessarily be reflective of the bulk solid. The structural information recovered from SC-XRD methods is sufficiently sensitive to minor differences in lattice spacings that can be used to distinguish elementally identical and structurally similar polymorphs. Therefore, the interpretation of single crystal diffraction is much less ambiguous than powder diffraction methods. Moreover, in the case of powder samples, the turbostratic (describing a crystal structure packing orientation) disorder of a layered material may alter the pattern, which could happen due to many reasons, including continuous diffraction rings and additional reflections that can occur due to superstructure ordering, resulting in ambiguities in the data interpretation. The surface area variation in the powder samples sometimes caused incommensurate modulation in the phase transitions due to the different measurement conditions. These are some of the obvious reasons as to why the PXRD patterns fail to match precisely with the simulated patterns. Herein, powder samples of co-crystals

(TAHOFN, TAHTFPN_G, TAHTFPN_O and TAHTCNB) were prepared from the soft grinding of single crystals without affecting the co-crystal structures and emission properties. Even under vigorous grinding of the afforded co-crystals, they exhibited unaltered emission behaviour. In the supplementary information figures (SI Fig 10-13), it can be clearly seen that experimental PXRD patterns of co-crystals are completely different to respective precursor's PXRD patterns, corroborating that co-crystals partners are not separated under external stimuli. This can also mean that applying stress and strain on the crystals may not translate well to describing co-crystals' purity from the given bulk properties of interest.

Comment 3. In Fig. 2d, cocrystal TAHTCNB starts to decompose around 100 °C, which is less stable compared to the other three cocrystals. However, authors claimed that TGA study reveals the formation of cocrystals and relatively high thermal stability of TAHTCNB (235 °C) rather than the other three cocrystals. Please explain.

Author Response: We thank the esteemed reviewer-1 for seeking this clarification. Yes, TAHTCNB co-crystal starts to decompose initially at around 70 °C (shown in Fig 2d, inset: red circle) and this degradation (less than 5 wt%) is due to the presence of solvent molecule in the crystal structure. However, actual solvent structure could not be observed in the single co-crystal structure, presumably this could be THF solvent (used for co-crystallization media). The presence of solvent molecule in the TAHTCNB crystal lattice can be noted at the solved single crystal structure (please see the attached check .cif data of solved crystal structure). However, actual decomposition temperature of the co-crystal was noticed to be round 235 °C which was also further confirmed from the melting point measurement of the co-crystals.

Fig. 2d. TGA graph for all the afforded four co-crystals and their corresponding decomposition temperatures.

No syntax errors found.
Please wait while processing

CIF dictionary
Interpreting this report

Datablock: TAHTCNB

Bond precision:	C-C = 0.0004 Å	Wavelength=0.71073
Cell:	a=8.1658(6) b=22.9815(16) c=22.0149(19)	
	alpha=90 beta=90 gamma=90	
Temperature:	293 K	
	Calculated	Reported
Volume	4131.4(5)	4131.4(6)
Space group	P n m a	P n m a
Hall group	-P 2ac 2n	-P 2ac 2n
Moiety formula	C34 H22, C10 H2 N4 [+ solvent]	C34 H22, C10 H2 N4
Sum formula	C44 H24 N4 [+ solvent]	C44 H24 N4
Mr	608.67	608.67
Dx, g cm ⁻³	0.979	0.979
Z	4	4
Mu (mm ⁻¹)	0.058	0.058
F000	1264.0	1264.5
F000'	1264.45	
h, k, lmax	9, 27, 26	9, 27, 26
Nref	3735	3732
Tmin, Tmax	0.983, 0.987	
Tmin'	0.983	
Correction method=	Not given	
Data completeness=	0.999	Theta(max)= 25.000
R(reflections)=	0.1207(1689)	wR2(reflections)= 0.3792(3732)
S =	1.048	Npar= 217

The following ALERTS were generated. Each ALERT has the format
test-name_ALERT_alert-type_alert-level.
Click on the hyperlinks for more details of the test.

Alert level B

PLAT084_ALERT_3_B	High wR2 Value (i.e. > 0.25)	0.38 Report
PLAT230_ALERT_2_B	Hirshfeld Test Diff for C18 --C20 .	9.7 s.u.

Alert level C

DIFM02_ALERT_2_C	The minimum difference density is < -0.1*ZMAX*0.75
	_refine_diff_density_min given = -0.531
	Test value = -0.525

Comment 4. Figure 4i: why the lifetimes of the samples exhibited obvious different decay?

Author Response: *We thank the esteemed reviewer-1 for seeking this clarification. The fluorescence lifetime decay is an intrinsic molecular property and influenced by its environment or the presence of other interacting molecules. In different chemical structures, the electronic redistribution of electrons due to optical excitation leads in many cases to a different reactivity in the ground and excited states. Common excited state phenomenon is governed by the charge-transfer (in D-A type systems), redox (electron transfer) and acid-base (proton transfer) reactions. Depending on the rate of the excited-state reactions which is related to the original fluorescence lifetime, the observed decay time measured with a TCSPC spectrometer may be single- or multiexponential.*

Since the emission cross-section is proportional to the luminescence efficiency and inversely on the luminescence decay time, a shortening of decay time would increase the efficiency of emission. The luminescence decay pattern showed a correlation of the intensity of fast-relaxing component with the degree of CT. Yes, the fluorescence lifetime decays of our four co-crystals

are different because of their variation in the degree of charge transfer and PLQY. Since, the PLQY is related to the rate of radiative decay and follows the equations (R_1 and R_2).

$$\tau = \frac{1}{\sum_i k_i} \quad R_1$$

$$\varphi = \frac{k_f}{\sum_i k_i} \quad R_2$$

Where, τ is the lifetime, φ is the PLQY, and k_f is the rate of spontaneous radiative emission and the denominator is the sum of all rates of excited state decay.

Moreover, TAHTCNB co-crystal exhibited the obvious different decay than other three co-crystals which can be ascribed to the delocalization of the excited state in the cis-configuration and results in multi-exponential decay. At the same time fitting results showed first order kinetics with fast-relaxing component for TAHOFN, TAHTFPN_G, and TAHTFPN_O, while TAHTCNB showed third order kinetics with delayed component at room temperature.

Comment 5. For the lifetime results from the decay curves in Fig. 4i and Fig. 4l, the fitting results and fitting goodness should be provided.

Author Response: We thank the esteemed reviewer-1 for seeking this clarification. We have now included the fitting decay curves in the revised manuscript (Fig. 4i/4l now 5i/5l). In addition, the time intervals, χ^2 including fitting results are shown in supporting Tables S7-S13.

Fig. 5: (i) TRPL spectra in log-scale for all the co-crystals in the solid state. (l) Transient PL spectra at different temperatures (100-300K) for TAHTCNB.

Comment 6. Polymorphs and cocrystals as luminescent materials and wide applications are hot topics, to arouse broader readerships, some strongly related works on luminescent molecular polymorphs and cocrystals (Angew. Chem. Int. Ed. 2011, 50, 12483; ACS Appl. Mater. Interfaces, 2018, 10, 22703; Angew. Chem. Int. Ed. 2023, 62, e202217054) and AIE type cocrystals (ACS Central Science, 2020, 6, 1169; Mater. Horiz. 2014, 1, 46) could be compared and included as references.

Author Response: *We thank the esteemed reviewer-1 for notifying these important references. We have incorporated the mentioned references into the comparison table and also cited them into the ref. no. 6, 8, 10, 39 and 44 in the revised manuscript.*

Comment 7. Currently, authors conducted crystal structure analysis after describing the optical properties and mechanisms. I suggest authors first do the crystal structure analysis and then investigate the optical properties and mechanism of cocrystals, which may help readers better understand the relationship between structure and property.

Author Response: *We thank the esteemed reviewer-1 for this constructive suggestion. As per the esteemed reviewers suggestion, we have now rearranged the content of optical properties and crystal structure analysis.*

Comment 8. For the optical waveguide, the loss efficiency needs to be compared with other organic or hybrid systems in literatures (Sci. Bull. 2022, 67, 2076; Sci. China Chem. 2022, 65, 408).

Author Response: *We thank the esteemed reviewer-1 for this comment. We have prepared a list of previous reports, mainly focusing on efficient crystal and co-crystals design and optical performances and have included the mentioned literatures into the comparison list and incorporated in the supplementary information Table SI Table 24.*

Comment 9. The image resolution needs to be further improved.

Author Response: *We thank the esteemed reviewer for this constructive comment. We have improved and incorporated the high resolution images in the revised manuscript.*

Reviewer #2 (Remarks to the Author):

Comment: COMMENTS TO AUTHOR:

This manuscript reported a series of rationally designed color-tunable CT co-crystals developed by utilizing the non-planar twisted donor TAH and planar acceptors OFN, TFPN and TCNB. The cis-isomeric co-crystal exhibits TADF characteristics, and the applications in optical waveguide and cell imaging have been demonstrated. The overall study is very interesting and attractive. The manuscript is also well organized. Therefore, the manuscript is enough to meet the requirements for publication after addressing the following issue.

Author Response: *We thank the esteemed reviewer-2 for appreciating our work, providing important constructive feedback and recommending it for publication.*

We also thank for the valuable suggestions and clarifications mainly on the temperature dependent PL emission (TADF) and cell imaging part. We have carefully elaborated the suggestions that has helped us to improve the manuscript significantly.

Comment 1. For TADF mechanism, except for temperature-dependent emission decay curves, the temperature-dependent emission spectra should also be provided. Additionally, it is crucial to fit the delay curves, and we need to know the changes in the proportion of long and short lifetimes at different temperatures to determine whether it is TADF emission.

Author Response: We thank the esteemed reviewer-2 for providing this constructive comment. We have recorded the temperature-dependent emission spectra for TAHTCNB and incorporated into the revised SI Fig. 38a.

Supplementary Fig. 38. A & b) Temperature dependent delayed emission spectra and fitted delayed decay profile for TAHTCNB co-crystal in solid state at 100, 200, 250K and 300K, respectively. c) Steady state PL at RT and 77K. d) PL at RT and phosphorescence at 77K.

We appreciate this comment from esteemed reviewer-2. To understand the nature of delay curves it is crucial to fit the delay curves to know the changes in the proportion of long and short lifetimes at different temperatures. As per this suggestion, we have now provided the fitted temperature dependent delayed decay curves for TAHTCNB in Fig. 5l in the main MS. The fitting curves showed that the increase of delayed decay with the increment of temperature from 100 to 300K, confirms TADF behaviour. Fitting results are provided in the revised SI Tables 11-13.

To further verify the TADF characteristics for TAHTCNB, we have now performed additional experiments including PL spectra at RT and phosphorescence spectra at 77K for TAHTCNB. From their respective spectral onset we could calculate the experimental ΔE_{ST} value (0.01 eV) which is nearly similar to computational ΔE_{ST} value.

Comment 2. The four microcrystals have different full width at half maximum (FWHM), what is this related to?

Author Response: *We thank the esteemed reviewer for this comment. The steady-state emission spectrum of the crystalline powder of TAHOFN, TAHTFPN_G and TAHTFPN_O and TAHTCNB co-crystals shows different featureless peak with a maximum wavelength at 540, 545, 560 and 590 nm, respectively. At the same time all four co-crystals showed variation in full width half maximum (FWHM) values of 63, 58, 56 and 72 nm, respectively. This can be attributed due to their distinct through-space CT interactions between TAH donor and acceptors viz. OFN, TFPN and TCNB. Moreover, TAHTCNB showed broad, higher FWHM and is red shifted compared to the other three co-crystals, because of its additional delayed fluorescence with prompt emission. FWHM is a good measure of sharpness, narrower FWHM value and is better and beneficial for fast radiative decay. Moreover, the large FWHM value (72 nm) for TAHTCNB indicates the strong CT-active TADF behaviour.*

Comment 3. I cannot understand why co-crystals of TAHTFPN_G and TAHTCNB have higher cytotoxicity. More experimental data should support the author's claim that cyanide is increased in the co-crystals. Because when using co-crystal to stain cells, co-crystal will decompose? What is the mechanism by which co-crystals can be internalized into cells? Which organelle in the cell is being stained? And what are the fluorescent dots outside the cells in Figure 8e-f?

Author Response: *We thank the esteemed reviewer-2 for this very important comment on the cell imaging performances of the co-crystals.*

TAHTFPN_G and TAHTCNB showed higher cytotoxicity due to the presence of cyanide substitution in the acceptor counter parts of those co-crystals. Since, TAHTCNB is having four cyanide group (-CN) in acceptor TCNB and compared to only two -CN group in TFPN acceptor for TAHTFPN_G, thus TAHTCNB showed higher toxicity.

However, we completely agree with the esteemed reviewer's comment that it is necessary to know the presence and internalization of the co-crystals into the cell. Therefore, as per this suggestion we have performed the internalization mechanism studies.

The mechanism of the internalization of the co-crystals was studied by measuring the fluorescence intensity of the internalized molecules in the presence of endocytosis inhibitors. Uptake of the TAHOFN, TAHTFPN_G and TAHTCNB decreased significantly, suggesting the active and energy-dependent mechanism of the endocytosis. To inhibit uptake via clathrin and caveolin-mediated pathway, chlorpromazine and beta-methyl cyclodextrin were added to the cells before co-crystal incubation, respectively. Chlorpromazine addition significantly reduced

the uptake of TAHOFN (64.9%), and TAHTFPN (75%), while beta-methyl cyclodextrin reduced the internalization of all three co-crystals (TAHOFN=36.4%, TAHTFPN_G=43.7% and TAHTCNB = 50.3%). Similar to chlorpromazine, both amiloride and micropinocytosis inhibitor reduced the internalization of TAHOFN (55.9%) and TAHTFPN (54.6%). Uptake studies with TAHOFN and TAHTFPN co-crystals confirmed the major role of the clathrin and macropinocytosis-mediated endocytosis, while TAHTCNB was predominantly internalized via caveolin- mediated endocytosis.

As the co-crystals were taken up by the cells through the process of endocytosis, they underwent internalization and eventually localized within the cytoplasm. In the case of the co-crystals, this mechanism allowed them to be enclosed within endosomes. The presence of the co-crystals within the endosomes signifies their successful internalization by the cells and suggests that they are being processed by the cellular machinery.

The presence of fluorescent dots observed outside the cells indicates the presence of co-crystals that have not been internalized by the cells. These dots, which appear as small points of fluorescence, are likely the result of co-crystals that were not effectively washed away during the washing steps of the experimental procedure.

Supplementary Fig. 58. Cellular uptake and internalization mechanism study.

Comment 4. The abbreviations that appear for the first time in the manuscript need to be explained, such as PXRD, TGA, DSC, etc.

Author Response: We thank the esteemed reviewer for this constructive comment. We have now mentioned the needful abbreviations in the revised manuscript.

Comment 5. "What does 'UV OFF' mean, does it refer to afterglow emission or under daylight, please check carefully."

Author Response: *We thank the esteemed reviewer for seeking this clarification. Actually the photographs of the co-crystals were taken under both “day light” and “UV light (365 nm)”, thus we have modified the figure and information in the revised manuscript.*

Reviewer #3 (Remarks to the Author):

Comment: COMMENTS TO AUTHOR:

This article by Parameswar Krishnan Iyer and co-workers reports a series of organic co-crystals using twisted aromatic hydrocarbon donor and three diverse planar acceptors. They report color-tunable emission in aggregated state via variable packing and through-space charge-transfer interactions. By adjusting the strength of acceptors, structural transformation into hybrid stacking modes generates color-specific polymorphs, an unprecedented configurational cis-isomer with very high photoluminescence quantum yield. Moreover, the cis-isomeric co-crystal exhibits triplet-harvesting thermally activated delayed fluorescence characteristics, and the 1D co-crystal exhibits efficient photon-transducing optical waveguides. Lastly, bright cell imaging performances were also obtained. However, color-tunable emission in aggregated state, color-specific polymorphs, efficient photon-transducing optical waveguides and bright cell imaging performances have ever been reported for other organic cocrystals. In a word, the novelty of this work is not enough, many data and expressions are not accurately. It is recommended to submit to other journals.

Author Response: *We thank the esteemed reviewer-3 for appreciating our work, providing important and constructive feedback and for these valuable suggestions. We have carefully elaborated the suggestions that has helped us to improve the manuscript significantly.*

The multiple criteria of novelty that we observed in this work (and also highlighted by the esteemed reviewer) includes the advancement of twisted hydrocarbon (TAH) based co-crystal. Careful mechanistic investigation of variable through-space charge-transfer collectively resulted in colour-tunable solid-state emission, high PLQY, color-specific polymorphism, configurational isomerism, thermally activated delayed fluorescence (TADF), optical waveguiding behaviour including bright cell imaging performances with promising results that have been achieved for the first time in the field of co-crystalline materials and demonstrated exclusively through this work.

The combined effects of multiple exceptional features in TSCT co-crystals in single platform using twisted aromatic hydrocarbon that is devoid of hetero atoms is not reported in literature. This research proves several unique milestones and has provided impetus to novel twisted TAH based co-crystals/polymorphs design strategies with unusual optical performances for exceptionally multi-active luminescent dynamic TSCT-co-crystals. Thus, the design of such novel luminescent co-crystals are the future of advanced optoelectronic materials development, and one of the major growing areas involving chemistry and material science aspects with immense application potential.

Therefore, such multi-functional twisted donor TAH based through-space charge-transfer induced photonic co-crystals design strategy have never been explored previously or conceived

to describe the fundamental aspects of structure-property correlations in depth, which is a very strong reason for us to submit this work for publication in Nature Communications.

Comment 1. There are lots of Grammatical errors :

a) “Remarkably, by adjusting the strength of acceptors, structural transformation into hybrid stacking modes generates color-specific polymorphs, an unprecedented configurational cis-isomer with very high photoluminescence quantum yield.” “To overcome this issues twisted aromatic hydrocarbons (TAHs) can play a crucial role,...” “As expected TA spectra depicted a positive signal near 600 nm while life-time was found to be higher than that of 77K (Fig. 4l). In addition PL spectra and TRPL decay profile of TAHTCNB co-crystal under inert/air and variable temperatures”, in which connectives (i.e., and) and comma should be added in suitable locations.

b) There are many other kinds of ungrammatical sentences, for example, “Hence, rational design of TAH-based multifunctional co-crystals with improved luminescence in aggregated/solid states, promising photon harvesting characteristics is of great significance which remains an imperative research area so far”, “Co-crystals with uniform chain-like molecular arrangements through continuous π - π interactions, their smooth surface and highly rigid crystalline structures, benefited to rationalize the unidirectional photon propagation at very low optical loss coefficient.”, and etc.

c) Lots of words were misspelled. For example, the word of col-or-tunable in “...with their respective tunable packing oriented col-or-tunable polymorphs and configurational isomeric binary assembly structures”, transducrive in “...that are transducrive to near-IR frequencies where most of the current growing areas involving fiber-optic-based communications are conducted”, co-crystalization in “color-specific polymorphic co-crystals were obtained by controlling the temperature during co-crystalization such as at room temperature (RT)...”, color-tunablity in “aggregation induced emission, color-tunablity and self-guiding singlet-triplet optical waveguides and biological application for the very first time.”, co-crystaliiation in “... investigate more critically the dissimilarity in CT/packing interactions and mechanistically determine the process of co-crystaliiation.”, Estima-tion in “Estima-tion of the optical loss coefficient (α) from the plot of I_{tip}/I_{body} versus the light propagation distance.”, and so on.

Please check carefully throughout the manuscript.

Author Response: *We thank the esteemed reviewer-3 for providing these constructive comments. We apologize for the Grammatical and typing errors. As per suggestions we have modified the errors and carefully revised the manuscript.*

Comment 2. In page 3, line 88, “All the integrated CT-complexes displayed different luminescence behaviour in the presence of different strength of acceptors as shown in Scheme 1.”, while only the CT mechanism for TAHTFPN_G and TAHTFPN_O were provided, and the CT character of TAHOFN was not proved.

Author Response: We thank the esteemed reviewer-3 for providing this constructive comment. Unlike experimental results, to understand the CT nature for all the co-crystals, NTOs distribution of lowest singlet and triplet states were calculated from their respective TD-DFT geometry and incorporated in the revised supplementary information figures (SI Fig.43-46). In particular, the NTOs of the lowest singlet state (S_1) for TAHOFN clearly showed a hybrid FMOs distribution, combination of locally excited (LE) and CT in nature. Due to weak acceptor strength of OFN, TAHOFN exhibited weak TSCT. Whereas, TAHTFP_O and TAHTCNB co-crystals exhibited a complete separation of FMOs in the S_1 , depicting stronger TSCT and therefore they exhibited red-shifted absorption and emission behavior.

Supplementary Fig. 43. Natural transition orbitals (NTOs) for lowest excited singlet and triplet excited states in TAHOFN.

Supplementary Fig. 44. Natural transition orbitals (NTOs) for lowest excited singlet and triplet excited states in TAHTFPN_G.

Supplementary Fig. 45. Natural transition orbitals (NTOs) for lowest excited singlet and triplet excited states in TAHTFPN_O.

Supplementary Fig. 46. Natural transition orbitals (NTOs) for lowest excited singlet and triplet excited states in TAHTCNB.

Comment 3. In page 4, line 109, “Besides, TAH co-crystals exhibited unusual AIEE phenomenon with superior water dispersibility, endowing good cellular internalization with bright cell imaging performances.” It is very difficult to understand that the co-crystals without any hydrophilic functionalization can present superior water dispersibility.

Author Response: *We understand the concern of esteemed reviewer-3 about the water dispersibility of TAH co-crystals. Typically, encapsulating AIEgen into nanoparticles (NPs) using amphipathic polymers as the matrix endows them with improved water dispersibility and biocompatibility, which allows them to be applied in both imaging and therapeutic applications. However, the design of encapsulation-free bio-probe for bioimaging studies remains challenging. However, we investigated that our integrated AIEgenic co-crystals*

exhibiting the formation of uniform nano-aggregates in water/DMSO mixture (99%), therefore, cells were treated with very dilute concentrations (10 μM) without encapsulation of TAHOFN, TAHTFPN_G and TAHTCNB one time and incubated for 6 h.

Furthermore to confirm the nano-aggregation formations, we have recorded the TEM images, DLS and zeta potentials of the dilute concentration of aggregated co-crystals and incorporated in the revised SI Fig. 30 and SI Fig. 31-33.

Supplementary Fig. 30. FETEM images of nano aggregates of TAHOFN, TAHTFPN_G and TAHTCNB co-crystals, respectively. (Inset: DLS size distribution graph)

Supplementary Fig. 31: Zeta-potential recorded in H₂O for TAHOFN at 10⁻⁵ M concentration.

Supplementary Fig. 32: Zeta-potential recorded in H₂O for TAHTFPN_G at 10⁻⁵ M concentration.

Supplementary Fig. 33: Zeta-potential recorded in H₂O for TAHTCNB at 10⁻⁵ M concentration.

Comment 4. In page 5, line 120, “Four color-tunable fluorescent rod-shaped single-crystalline co-crystals denoted as TAHOFN, TAHTFPN_G, TAHTFPN_O and TAHTCNB were prepared via solvent diffusion method, by using 1:1 stoichiometry ratio of the precursor molecules in THF solvent.”, in which the solvent diffusion method is not correct, because the authors adopted evaporation method to prepare all the co-crystals.

Author Response: *We thank the esteemed reviewer for this comment and agree with the esteemed reviewers' statement. We have modified now to solvent evaporation method for all of the co-crystals preparation.*

Comment 5. In page 5, line 125, “TAH co-assembled very promptly (within 48 hours) and resulted in rod-like four distinct colored CT complexes.” It should be better to express clearly which acceptor did TAH co-assemble with.

Author Response: *We thank the esteemed reviewer-3 for seeking this clarification. TAH promptly co-assembled with the selected acceptors viz. OFN, TFPN and TCNB (within 48 hours), resulting in rod-like four distinct colored CT complexes.*

Comment 6. In page 5, line 126 to 128, “Notably, color-specific polymorphic co-crystals were obtained by controlling the temperature during co-crystallization such as at room temperature (RT) for TAHTFPN_G and at 60°C for TAHTFPN_O, while others were grown at RT (Fig. 1a).” Is 60°C the temperature during dissolving? Why do the author only provided the case of TAHTFPN at 60°C? How about TAHOFN and TAHTCNB?

Author Response: *We thank the esteemed reviewer-3 for asking this clarification. Yes, the color-specific polymorphic co-crystals of TAHTFPN were obtained by controlling the temperature during dissolving the mixture. TAHTFPN_G was obtained at room temperature (RT), whereas TAHTFPN_O at 60 °C. In contrast, others co-crystals were grown at RT and no other polymorphs or other structures were observed under temperature variation.*

Comment 7. In the SI file, “Supplementary Fig. 1. (a) Fluorescence microscopy images for TAH, TAHOFN, TAHTFPN_G, TAHTFPN_O and TAHTCNB co-crystals with the excitation wavelength 405 nm. (b) While at the bottom images are the bright field images.”, the caption includes TAH, but the data of TAH cannot be found.

Author Response: *We thank the esteemed reviewer-3 for the careful observation. We apologize for this misleading information, Supplementary Fig. 1 actually contains the FM images of TAHOFN, TAHTFPN_G, TAHTFPN_O and TAHTCNB co-crystals.*

Comment 8. In page 7, Fig. 2e., ¹H NMR data collected using CDCl₃ as the solvent. Actually, cocrystals are not stable in organic solvents, which can be well dissolved and the corresponding donor and acceptor exist in isolated single molecular state. Therefore, using liquid ¹H NMR to characterize the cocrystals is not a correct tool.

Author Response: *We thank the esteemed reviewer-3 for this insightful comment and completely agree that in the solution, donor and acceptor exists in isolated single molecular state. Moreover, in the present system, by recording ¹H NMR using CDCl₃ solution, we could still notice the upfield and downfield chemical shift, corresponding to the actively participating*

hydrogen atom (present in TAH) for the weak non-covalent bonding interactions with different acceptors. Besides, the possibility of conformational change in case of TAHTCNB (cis geometry) under the influence of TCNB lead to shielding (upfield chemical shift) in TAH rather trans geometry of TAH in TAHOFN and TAHTFPN system. These observation clearly depicted that presence of foreign entity such as acceptors, TAH is a highly efficient donor to co-assemble with acceptors spontaneously. By redistributing its electron clouds in presence of different electronegative atoms in the acceptors via electrostatic Coulomb interactions which directly influenced the chemical shift of TAH proton even in the solution state.

Besides, we also recorded ^1H NMR in the aggregated form using DMSO- d_6 and D_2O mixture (SI Fig. 19), however we could see similar peak shifting to CDCl_3 , yet very weak peak intensity and strong solvent peaks were noted which could be due to the poor proton coupling between solvent and solute (co-crystal aggregates). The rapid formation of co-crystal aggregates during ^1H NMR measurements hinders to collect the high intensity peaks. However, our main emphasis is to understand the influence of weak non-bonding interactions between guest acceptors and TAH donor that resulted in chemical shifting in the TAH in solution or aggregates in presence of different strength of electron withdrawing atoms. We assume that different electro negative environment even in liquid phase can affect the chemical shift. We can successfully observed that, indeed there is a slight changes in the chemical shifts. For precise characterization solid-state NMR spectroscopy may be the correct tool to avoid such conflict, however due to lack of the instrument facility we could not record solid-state NMR.

Supplementary Fig. 19. (e) Stacked ^1H NMR spectra for TAHOFN, TAHTFPN and TAHTCNB co-crystals, recorded in DMSO- d_6 and D_2O mixture at 298K. (Inset: images of NMR tubes containing co-crystals taken under 365 nm UV lamp)

Comment 9. In page 9, Fig. 3., the TEM images of the cocrystals are presented, but the detailed preparation procedures of the TEM samples cannot be found. It should be better to provide it.

Author Response: *We thank the esteemed reviewer-3 for this constructive comment. For TEM images, the samples were prepared using 5 μ L of the sample solution which was drop-casted on carbon coated copper grid (300 mesh Cu grid with thick carbon film from Pacific Grid Tech, USA) and allowed to air dry for 2 minutes and then the excess sample was soaked up carefully with a tissue paper. The grid was then immediately freeze-dried and the FETEM images were recorded in JEOL JEM-2100F microscope. This sample preparation details are included in the revised manuscript.*

Comment 10. In page 9, Fig. 3e., the TEM images of TAHOFN give a helical ribbon-like morphology rather than rod, which is not consistent with the results of the SEM data and AFM data.

Author Response: *We thank the esteemed reviewer-3 for this constructive comment and agree that the TEM images of TAHOFN looks like helical ribbon-like morphology which is not consistent with the results of the SEM data and AFM data. This is because the highly energetic electron beam irradiation to TAHOFN, destroying the crystal structures thereby turning those to amorphous domains and hence TAHOFN changes its morphology (Micron **2004**, 35, 399–409). Also, no crystal lattice was observed in selected area electron diffraction (SAED) analysis (inset Figure 3e). Since, in the SEM and AFM, the electron beam only scans the surface of the sample, but in TEM, the electron beam passes through the sample and therefore a partial damage transformed the actual rod like morphology to helical ribbon-like morphology.*

Comment 11. In page 10, line 234, “To investigate the luminescent features of the afforded co-crystals, UV-visible and photoluminescence (PL) studies were performed in solid crystals and aggregated state respectively.” Actually, solid crystals are also aggregated state, so the “solid crystals and aggregated state” here can be replaced by “bulk crystals and micro/nano-scale crystals”.

Author Response: *We thank the esteemed reviewer-3 for this constructive comment and agree that the solid crystals are also aggregated state, hence the above “solid crystals and aggregated state”, has been replaced to “bulk crystals and micro/nano-scale crystals” in the revised manuscript.*

Comment 12. In page 10, line 236 to 249, “TAHOFN, TAHTFPN_G exhibited blue-shifted absorption at 460 and 430 nm whereas TAHTFPN_O and TAHTCNB showed a red-shifted absorption at 490 and 500 nm related to the absorption of parent TAH (485 nm), deciphering TSCT interactions between constituent precursors.” If TAHOFN with a blue-shifted absorption really includes the TSCT interaction? The author only provided the corresponding data about TSCT interaction for the other three cocrystals, i.g., “FMOs distribution suggested that HOMO

of the three co-assemblies were mainly located on the donor TAH, whereas LUMO aligned with the acceptors TFPN and TCNB, leading to well separated HOMO and LUMO, clearly indicating the TSCT characteristics of the co-assemblies (Supplementary Fig. 32-33)38.”

Author Response: *We thank the esteemed reviewer-3 for this constructive comment. Yes TAHOFN exhibited weak TSCT behavior as compared to other three co-crystals. To confirm this, we have now performed the NTOs for all the co-crystals and incorporated in the revised supplementary information figures (SI Fig. 43-46). The NTOs of TAHOFN clearly showed that the lowest singlet state (S_1) is a hybrid FMOs distribution, and a combination of locally excited (LE) and CT in nature. Therefore, it exhibited weak TSCT. Whereas, in TAHTFP_O and TAHTCNB co-crystals, the complete separation of FMOs is S_1 , depicting stronger TSCT and therefore red-shifted absorption and emission behavior. Therefore, from the computational calculations of the afforded CT-complexes and single-crystal X-ray analysis it was clearly demonstrated that the electron clouds in conformers are redistributed, including electrostatic Coulomb interactions, which directly influenced the degree of CT, corroborating structure-packing property relationship in the co-crystals.*

Supplementary Fig. 43. Natural transition orbitals (NTOs)³ for lowest excited singlet and triplet excited states in TAHOFN.

Comment 13. In page 10, line 238, the full name of TSCT should be provided here because of its first occurrence.

Author Response: *We thank the esteemed reviewer-3 for this comment. We have updated the full name as through-space charge transfer (TSCT).*

Comment 14. In page 10, line 247 and 248, “Therefore, the TAH based co-assembly approach reduces aggregation caused quenching effect and showed improved condensed state emission^{10,33}.” There is no data that prove ACQ effect of TAH. It should be better to

supplement the corresponding data of TAH solution and pure TAH solids, including the FM images, quantum yield, life time, and etc.

Author Response: We thank the esteemed reviewer-3 for this constructive comment. In this work, the precursor donor TAH is an AIE active chromophore. Moreover, in our previous report, the ACQ effect of developed charge-transfer complex of TAHTFQ and TAHTCQ, using TAH and quinone derivatives have been explored and discussed in detail the photophysical properties of TAH (Langmuir 2021, 37, 8024–8036, cited in the ref 42). However, to correlate the current findings, we could reproduce solid and aggregated state properties for TAH and incorporated in the revised supplementary information (SI Fig. 29) for the easy co-relation with the present investigations.

Supplementary Fig. 41. (a) Photoluminescence spectra for solid pristine TAH. (b) Aggregation–induced emission at different fractions of THF and water. (c) Relative intensity at various fractions of THF/water (inset aggregation images taken under 365 nm UV lamp and absolute PLQY). (d) TRPL decay in THF and water. (e–g) FESEM, FETEM and fluorescence images of TAH nano-ribbons formed in water.

Comment 15. In page 14, in the caption of Fig. 5., “(a_i, b_i, c_i and d_i)”, “(a_ii, b_ii, c_ii and d_ii)” and “(a_iii, b_iii, c_iii and d_iii)” should be corrected into “(a-i, b-i, c-i and d-i)”, “(a-ii, b-ii, c-ii and d-ii)” and “(a-iii, b-iii, c-iii and d-iii)”.

Author Response: We thank the esteemed reviewer-3 for this constructive comment. We have now modified the Figure label in the caption and main text as well.

Comment 16. In page 15, line 348-349, the expression of “In order to understand the red-shifted CT-emission mechanism for all four co-crystals accurately, the degree of TSCT (ρ) was calculated” is not correct because only two cocrystals presented red-shifted CT-emission.

Author Response: *We thank the esteemed reviewer-3 for this comment. We have now modified the sentences as follows: “In order to understand the variable CT-emission mechanism accurately, the degree of TSCT (ρ) was calculated for all four co-crystals”.*

Comment 17. In page 20, line 469-470, the authors describe “As shown in Fig. 8a-c, we did not observe significant dark toxicity of TAHOFN, even up to 100 μ M concentration.” I don’t think so, as showed in Fig. 8a, the cell viability is already lower than 80% even under the concentration of 30 μ M, indicating high dark toxicity, which is also proved by Fig. 8d-f.

Author Response: *We thank the esteemed reviewer-3 for this careful observation. We agree with the statement. We apologize for this misleading information. We have now updated the information in the revised manuscript as follows: “As shown in Fig. 8a-c, we observed dose-dependent toxicity of TAHOFN and TAHTFPN_G. From the viability calculations, it was found that TAHTFPN was more cytotoxic than TAHOFN”.*

Comment 18. In page 21, line 479, “Microscopy-based uptake study was carried out to confirm the internalization of PSs molecules inside the cells.” The full name should be provided here.

Author Response: *We thank the esteemed reviewer-3 for this constructive comment. For fluorescence based imaging we used confocal microscope. This information is updated in the revised manuscript.*

Comment 19. In page 21, in Fig. 8d-f, because all the nucleus under the three cases were stained with PI, they should present the similar color. Why do the case of TAHTCNB is blue and different from the other two cases?

Author Response: *We thank the esteemed reviewer-3 for this important comment. We apologize for this incomplete information. Actually, in the confocal microscope imaging study, the MCF-7 cells were incubated with the co-crystals of TAHOFN, TAHTFPN_G and counterstained with propidium iodide (PI). Whereas, in the case of TAHTCNB we used 4',6-diamidino-2-phenylindole (DAPI). We have now update this information in the revised manuscript.*

Comment 20. In page 25, line 630, there is an extra “of” in the sentence of “Following the attachment, cells were treated with 10 μ m of of TAHOFN, TAHTFPN_G and TAHTCNB one time and incubated for 6 h.”

Author Response: *We apologize for this typing error and thank the esteemed reviewer-3 for careful observation. We have removed the extra “of” from this sentence in the revised manuscript.*

Comment 21. As we all know, the materials applied in cell imaging should be nanoscale, so how did the authors prepare the samples used in cell imaging should be detailed.

Author Response: *We thank the esteemed reviewer-3 for this constructive comment. We have included the complete method for the sample preparation for cell imaging studies. We have included in the revised manuscript.*

To evaluate the imaging efficiency of TAHOFN, TAHTFPN_G, and TAHTCNB inside the cellular environment, MCF-7 cells were seeded on 96-well plates at a density of 6000 cells per well. TAHOFN, TAHTFPN_G, and TAHTCNB were dissolved in DMSO at a concentration of 10 mM. The co-crystals were then dissolved in DMEM medium to achieve a concentration of 10 μ M for addition to the cells. After 6-hours incubation, the cells were washed with PBS three times and fixed with 4% formaldehyde. Images were captured using a confocal microscope with 488 nm and 561 nm lasers as excitation sources.

CHANGES MADE IN THE REVISED MANUSCRIPT:

1. Three supplementary videos (SI videos 1, 2 & 3) are included for rapid co-crystallization using solid-state grinding method
2. Fig. 2d is modified, where red dotted boxes in the inset of d is inserted to describe the TAHTCNB decomposition temperature followed by solvent decomposition.
3. Fig. 6 shifted to Fig. 4. Thus Fig. 4 is now Fig 5 and Fig 5 changes to Fig. 6 in the revised manuscript, related texts are arranged accordingly.
4. Revised - Fig. 5i and 5l are modified, where fitting, fitting goodness and average lifetime values are mentioned in the decay curve and inset.
5. Design principle is included in the “Co-crystals design and formulation” subsection under Result Discussion section.
6. Lowest singlet and triplet NTOs of all four co-crystal are provided in the “Intrinsic mechanism for color-tunable emission”
7. Cellular internalization studies of co-crystals are incorporated in the revised manuscript.
8. All the full form of the abbreviations are provided in the revised manuscript.
9. Measurement details of temperature dependent delayed emission and lifetime, sample preparation for TEM images, sample preparation for cell imaging and cellular internalization methods are included in the “Measurements” sub-section in Method section in revised manuscript.
10. Supplementary information- “Preparation of Co-crystals” section is shifted to main revised manuscript.
11. Seven new citations (ref 6, 8, 10, 39, 42, 44 and 59) are included in the revised manuscript.
12. Supplementary information- Fig. 19 is included, where stacked ^1H NMR spectra for all the co-crystals are recorded in DMSO-d₆ and D₂O mixture at 298K.
13. Supplementary information- Fig. 29 is included, where photophysical and morphological characterization of TAH donor is described.
14. Supplementary information- Fig. 30-33 are included, where FETEM images (inset: DLS graph for size distribution) and zeta-potential of co-crystal nano aggregates formed in DMSO/H₂O (99%, f_w) is incorporated.
15. Supplementary information- Fig. 38 is included, where, temperature dependent delayed emission spectra for TAHTCNB co-crystal in solid state, PL at RT and phosphorescence at 77K, steady state PL at RT and 77K for TAHTCNB solid state (powder) are incorporated.

16. Supplementary information- Table 11-13 are included, where fitting results for delayed decay at 300, 200 and 100K for TAHTCNB co-crystal is incorporated.
17. Supplementary information- Fig. 43-46 are included, where, natural transition orbitals (NTOs) for lowest excited singlet and triplet excited states for all four co-crystals are incorporated.
18. Supplementary information- Fig. 58 is included, where, cellular internalization mechanism studies using co-crystals are incorporated.
19. A modified Supplementary Table 24 is included, where, a brief summary of the previous report on various photonic crystals an optically waveguide active molecular crystals performances are summarized
20. All the modified text Figures and included discussions are highlighted in yellow background in the revised manuscript.

REVIEWERS' COMMENTS

Reviewer #1 (Remarks to the Author):

In my view, the authors have answered all the questions and revised related points, and thus this revised work can be published as it is.

Reviewer #2 (Remarks to the Author):

[Note from the editor: Please also see attached PDF]

The authors have done additional work to address the reviewers' comments and this revised manuscript is now stronger than the past version of the manuscript. The authors have satisfactorily addressed my past reviewer comments and I think that this version of the manuscript can now be published.

Reviewer #3 (Remarks to the Author):

it can be accepted now

Response to Reviewer's and Editorial Comments for Manuscript ID: NCOMMS-23-09657A

Reviewer(s)' Comments to Author:

Reviewer #1 (Remarks to the Author):

Comment: In my view, the authors have answered all the questions and revised related points, and thus this revised work can be published as it is.:

Author Response: *We would like to thank esteemed reviewer-1 for the helpful suggestions and the recommendation to publish this manuscript in Nature Communications.*

Reviewer #2 (Remarks to the Author):

Comment: The authors have done additional work to address the reviewers' comments and this revised manuscript is now stronger than the past version of the manuscript. The authors have satisfactorily addressed my past reviewer comments and I think that this version of the manuscript can now be published.

Author Response: *We would like to thank the esteemed reviewer-2 for appreciating the revision work, providing positive feedback and finding suitable for its publication in Nature Communications.*

Reviewer #3 (Remarks to the Author):

Comment: it can be accepted now

Author Response: *We would like to thank esteemed reviewer-3 for your helpful suggestions, constructive feedback and positive comments to polish our paper in Nature Communications.*

Editor(s)' Comments to Author:

Comment: Your manuscript entitled "Highly Efficient Color-Tunable Organic Co-crystals: Unveiling Polymorphism, Isomerism, Delayed Fluorescence for Optical Waveguides and Cell-imaging" has now been seen again by our referees, whose comments appear below. In light of their advice I am delighted to say that we are happy, in principle, to publish a suitably revised version in Nature Communications under the open access CC BY license (Creative Commons Attribution 4.0 International License).

We therefore invite you to revise your paper one last time to address the remaining concerns of our reviewers and our editorial requests in the attached document(s). At the same time we ask that you edit your manuscript to comply with our policies and formatting requirements and to maximise the accessibility and therefore the impact of your work.

Please see the attached document(s), listing a number of points that must be addressed. Failure to comply with our editorial requests will cause delays in accepting your manuscript. Please also see the Nature Communications formatting instructions for further information.

Author Response: *We would like to thank esteemed Editor for the advice on improving the manuscript, and to finally publish this paper in Nature Communications. All the points have been addressed very carefully.*

The author checklist file is also being uploaded with all the responses listed in it.